# The *C. elegans* gonadal sheath Sh1 cells extend asymmetrically over a differentiating germ cell population in the proliferative zone

Xin Li, Noor Singh, Camille Miller, India Washington, Bintou Sosseh, Kacy Lynn Gordon*

Department of Biology, The University of North Carolina at Chapel Hill, Chapel Hill, United States

**Abstract** The *Caenorhabditis elegans* adult hermaphrodite germline is surrounded by a thin tube formed by somatic sheath cells that support germ cells as they mature from the stem-like mitotic state through meiosis, gametogenesis, and ovulation. Recently, we discovered that the distal Sh1 sheath cells associate with mitotic germ cells as they exit the niche Gordon et al., 2020. Here, we report that these sheath-associated germ cells differentiate first in animals with temperature-sensitive mutations affecting germ cell state, and stem-like germ cells are maintained distal to the Sh1 boundary. We analyze several markers of the distal sheath, which is best visualized with endogenously tagged membrane proteins, as overexpressed fluorescent proteins fail to localize to distal membrane processes and can cause gonad morphology defects. However, such reagents with highly variable expression can be used to determine the relative positions of the two Sh1 cells, one of which often extends further distal than the other.

*For correspondence:
kacy.gordon@unc.edu

Competing interest: The authors declare that no competing interests exist.

## Editor's evaluation

This work extends the previous findings by the authors that suggested the lack of 'bare region' in the *C. elegans* gonad, which was previously postulated to exist between germ cells that are encapsulated by the distal tip cell and those that are encapsulated by sheath cells. The authors addressed the concerns posed by Tolkin et al. that proposed that the bare region does exist. However, discrepancies remain between the current manuscript and the manuscript by Tolkin et al., which should be resolved in the field in the future. Overall, the work presented here is important and of broad interest as it concerns the regulation of the stem cell niche and how cells that are destined to differentiate exit the niche and proceed to differentiation by interacting with the stromal cells.

## Introduction

The *Caenorhabditis elegans* hermaphrodite gonad is a fruitful system in which to study organogenesis, meiosis, and stem cell niche biology. Recent work from our group (*Gordon et al., 2020*) used two endogenously tagged alleles of genetically redundant innexin genes *inx-8* and *inx-9* to visualize the somatic gonadal sheath of the *C. elegans* hermaphrodite. We discovered that the distal most pair of sheath cells, called Sh1, lies immediately adjacent to the distal tip cell (DTC), which is the stem cell niche of the germline stem cells. Previously (based on electron microscopy and on cytoplasmic GFP overexpression from transgenes active in the sheath [*lim-7p::GFP*] [*Hall et al., 1999*] or its progenitor cells [*lag-2p::GFP*] [*Killian and Hubbard, 2005*]), Sh1 cells were thought to associate only with germ

cells well into the meiotic cell cycle, so our finding required a reimagining of the anatomy of the distal gonad.

Here, we confirm that the Sh1 cells fall at the boundary of a population of germ cells in a stem-like state, report other markers that label the Sh1 cells, and verify that these markers can be used to assess gonad anatomy without unduly impacting the gonad itself. We also discuss reagents that are not suitable markers of Sh1 cells, including an overexpressed, functional cell death receptor that is used to mark Sh1 in a recent study (*Tolkin et al., 2022*). Finally, we consider best practices for using endogenously tagged proteins for cell and developmental studies.

## Results
### Distal Sh1 associates with the population of germ cells that differentiate first when progression through mitosis is halted or Notch signaling is lost

Our first experiment addresses in a new way the question of what type of germ cells associate with the distal Sh1 cell. The DTC expresses the Notch ligand LAG-2, which is necessary to maintain the germline stem cell pool (*Henderson et al., 1994*). Recent work has shown that the active transcription of Notch targets *sygl-1* and *lst-1* (*Lee et al., 2019*) and the accumulation of their proteins (*Shin et al., 2017*) is restricted to the distal-most germ cells, describing a population of stem-like germ cells ~6–8 germ cell diameters (~25–35 µm) from the distal end of the gonad. Our recent work (*Gordon et al., 2020*) reported that the position of Sh1 coincides with *sygl-1* promoter's activation boundary on one side and the accumulation of the meiotic entry protein GLD-1 on the other, consistent with the hypothesis that the distal edge of Sh1 falls at the boundary of that stem-like cell population, ~30 µm from the distal end of the gonad.

A similarly positioned stem-like germ cell population was found in earlier work that used temperature-sensitive alleles to perturb germ cell fate or progress through the cell cycle (*Cinquin et al., 2010*). The readout was germ cell fate as determined in one of two ways. Anti-phosphohistone H3 staining of proliferative cells and GLD-1 antibody costaining for cells accumulating meiotic factors shows where cells are dividing and beginning to differentiate, respectively. Alternatively, the 'transition zone' in germ cell nuclear morphology between 'mitotic' and 'meiotic' zones can be visualized by the presence of crescent-shaped nuclei of meiotic prophase observed by DAPI staining. We used this latter method of visualizing nuclear morphology.

We repeated these experiments in strains that have tagged innexins to mark the distal sheath Sh1 cells to ask which population(s) of germ cells are associated with Sh1. Here, we describe the original findings and their interpretations, and then our new findings. First, an *emb-30* temperature-sensitive allele is known to cause germ cell division to arrest at the metaphase-anaphase transition, thus halting the distal-to-proximal movement of germ cells that is driven by the proliferation of more distal cells (*Cinquin et al., 2010*). In a wild-type gonad, germ cells differentiate (enter and progress through the meiotic cell cycle) as they move from distal to proximal (*Figure 1A*, left). In *emb-30(ts)* gonads, a shift to the restrictive temperature causes proliferation to halt and germ cells to remain stationary within the gonad (*Figure 1A*, right). These cells can now differentiate in place—or remain in the undifferentiated state—depending on their exposure to the stemness cue. Germ cells that remain in the niche at the distal end of the gonad do not differentiate after 15 hr at the restrictive temperature, while more proximal germ cells do differentiate. Nuclear morphology differs between these two regions of the germline.

We hypothesized that the transition in nuclear morphology in *emb-30(tn377)* animals would shift proximally after 15 hr at the restrictive temperature (as had previously been observed), to ultimately fall at the distal position of the Sh1 cell as visualized by mKate::INX-8; we hypothesize that the position of the Sh1 cell would itself not be affected by the temperature shift. Indeed, this is what we found (*Figure 1B and E*), supporting our prior conclusion that there is germ cell fate asymmetry across the DTC-Sh1 boundary. These results are independent of culture time of controls at the permissive temperature (*Figure 1—figure supplement 1*). The Sh1 cells cover proliferative germ cells outside the niche that are closer to differentiating than those under the DTC.

The second set of experiments using temperature-sensitive alleles to reveal differences in germ cell fate along the distal-proximal axis uses *glp-1(ts)* alleles to stop Notch signal transduction and

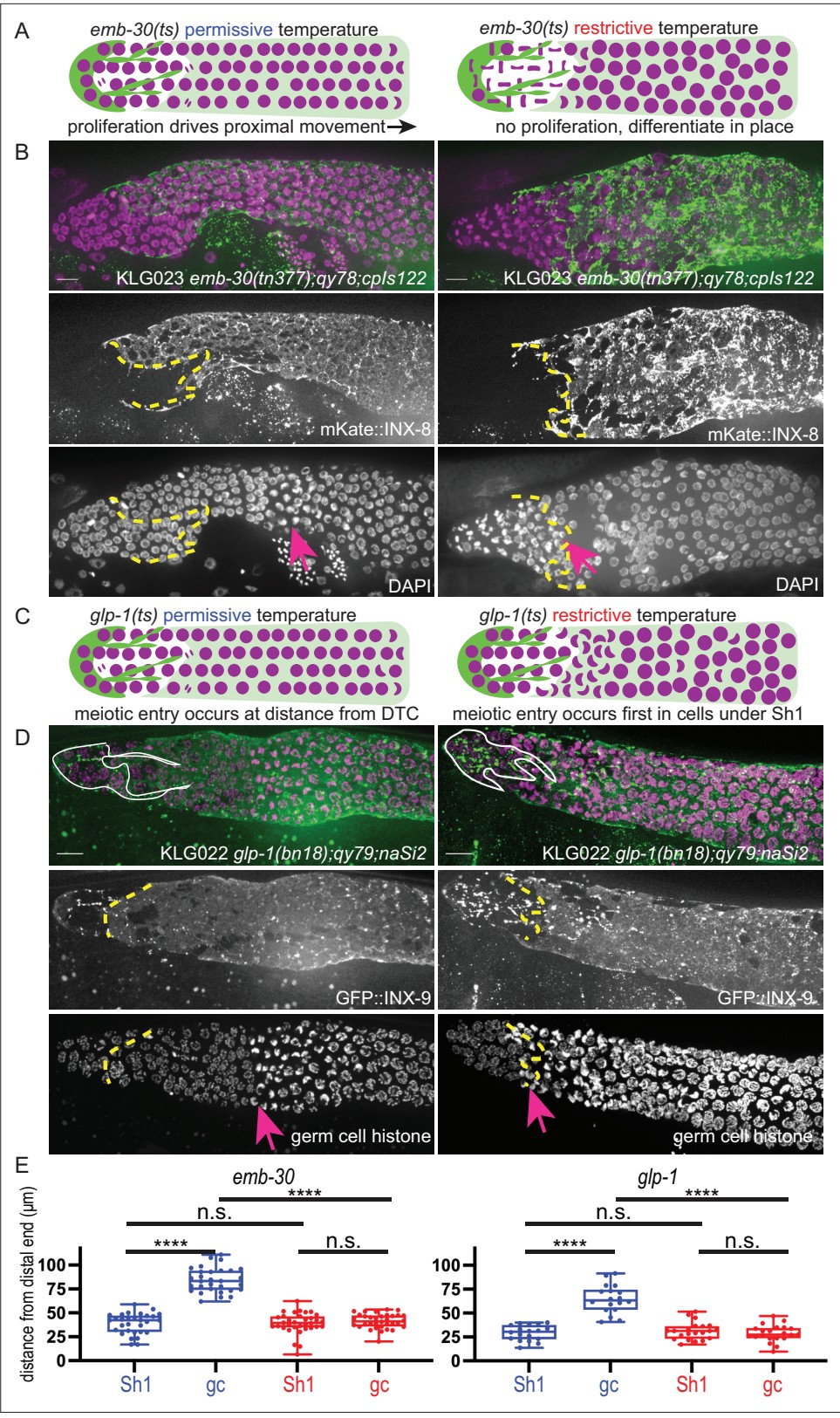

**Figure 1.** The Sh1 cells associate with proliferative germ cells that are on the path to differentiation. (**A**) Schematic of hypothesis for *emb-30(tn377)* experiment. Germ cell (gc) nuclei shown in magenta, somatic gonad cells shown in green (distal tip cell [DTC]), and transparent green (Sh1). (**B**) Gonads from KLG023 *emb-30(tn377);qy78;cpIs122* worms reared at permissive (left column) and restrictive (right column) temperatures. Top, merged image.

*Figure 1 continued on next page*

*Figure 1 continued*

Middle, mKate::INX-8 labeling Sh1 (edge outlined with yellow dashed line). Bottom, DAPI staining labeling all nuclei with pink arrow marking gc transition and same yellow dashed line as in middle image showing Sh1 edge. (**C**) Schematic of hypothesis for *glp-1(ts)* experiment. (**D**) Gonads from KLG022 *glp-1(bn18);qy79;naSi2* worms reared at permissive (left column) and restrictive (right column) temperatures. Top, merged image. Middle, GFP::INX-9 labeling DTC (outlined in white) and Sh1 (edge outlined with yellow dashed line). Bottom, germ cell histone mCherry (*naSi2[mex-5p::H2B::mCherry]*) with pink arrow showing gc transition and same yellow dashed line as in middle image showing Sh1 edge. Note that the *glp-1(bn18)* allele is not fully wild type at permissive temperatures and is known to have a shortened proliferative zone (**Fox and Schedl, 2015**). (**E**) Box plots overlaid with all datapoints measuring the distal position of Sh1 and the position of the transition zone in germ cell nuclear morphology. Permissive temperature shown in blue; restrictive temperature shown in red. Permissive *emb-30* N=30; restrictive *emb-30* N=34. Permissive *glp-1* N=18; restrictive *glp-1* N=21. A one-way ANOVA to assess the effect of temperature on proximodistal position of gonad features was performed, and was significant for *emb-30*: $F_{3,124}$=134.5, p<0.0001. Tukey's multiple comparison test found that the mean values of the positions of Sh1 and the germ cell transition zone were significantly different at the permissive temperature (mean difference of –45.25 μm, 95% CI –52.38 to –38.12 μm, p<0.0001), but not at the restrictive temperature (mean difference of –2.30 μm, 95% CI –9.00 to 4.40 μm, p=0.808). The position of the germ cell transition zone differed at the permissive vs. restrictive temperatures (mean difference of 42.87 μm, 95% CI 35.95 to 49.79 μm, p<0.0001), but the Sh1 position did not (mean difference of –0.078 μm, 95% CI –7.00 to 6.84 μm, p>0.9999). This pattern is observed across replicates and various controls (**Figure 1—figure supplement 1**). Similar results were obtained for *glp-1*: $F_{3,74}$=52.84, p<0.0001. Tukey's multiple comparison test found that the mean values of the positions of Sh1 and the germ cell transition zone were significantly different at the permissive temperature (mean difference of –35.51 μm, 95% CI –44.59 to –26.43 μm, p<0.0001) but not at the restrictive temperature (mean difference of 2.514 μm, 95% CI –5.892 to 10.92 μm, p=0.861). The position of the germ cell transition zone differed at permissive vs. restrictive temperatures (mean difference of 36.02 μm, 95% CI 27.27 to 44.77 μm, p<0.0001), but the Sh1 position did not (mean difference of –1.997 μm, 95% CI –10.75 to 6.753 μm, p=0.9318). All scale bars 10 μm.

The online version of this article includes the following source data and figure supplement(s) for figure 1:

**Source data 1.** Source data used to generate plots of distal sheath and germ cell transition zone measurements at permissive and restrictive temperatures for mutant strains shown in **Figure 1**.

**Figure supplement 1.** Robustness of *emb-30* temperature shift experimental results to timing of control population.

**Figure supplement 1—source data 1.** Source data used to generate plots of distal sheath and germ cell transition zone measurements at permissive and restrictive temperatures for mutant strains shown in **Figure 1— figure supplement 1**.

observe where and when germ cells acquire features of differentiation (**Figure 1C**). The same study (**Cinquin et al., 2010**) found that the *glp-1(q224)* temperature-sensitive allele reared at the restrictive temperature over a 9 hr time course showed a progressively shrinking mitotic zone (as assessed by nuclear morphology of DAPI stained gonads) until hour ~5.5, at which time the remaining distal most ~5 rows of germ cells differentiate as a pool. A subsequent study (**Fox and Schedl, 2015**) used a similar approach (but with the *glp-1(bn18)* temperature-sensitive allele, slightly different timing, and antibody staining to determine cell fate) and found a similar result, and additionally discovered that progress through the cell cycle influenced the precise timing of germ cell differentiation.

We used the *glp-1(bn18)* allele to deactivate Notch signaling in a strain with GFP::INX-9 to visualize the Sh1 cells and the fluorescent histone marker *naSi2(mex-5p::H2B::mCherry)* to visualize germ cell nuclei (**Figure 1D and E**). We hypothesized that after 6 hr at the restrictive temperature, only the distal-most pool of stem-like cells will not have taken on the crescent-shaped nuclear morphology of meiotic germ cells, while the germ cells under Sh1 will have entered the meiotic cell cycle. We predicted that the Sh1 cell would not change its position across this time interval. Indeed, this is what we found, further supporting our hypothesis that the Sh1-associated germ cells are closer to differentiation than those under the DTC, which are the last to differentiate.

Results from these temperature-sensitive mutants confirm what the markers of germ cell fate revealed in **Gordon et al., 2020**, which is that the Sh1 cell associates with germ cells in the proliferative zone that have left the stem cell niche and are on the path to differentiation, while the stem-like germ cells lie immediately distal to the Sh1 cell at its interface with the DTC.

## Different endogenously tagged membrane proteins reveal a distal position of Sh1

These experiments made use of the endogenously N-terminal tagged *inx-8(qy78[mKate::inx-8])* and *inx-9(qy79[GFP::inx-9])* alleles (*Figure 2A and B*) generated by *Gordon et al., 2020*. Both tagged proteins are highly specific for the somatic gonad throughout development; in the adult, their expression differentiates, with INX-8 signal diminishing from the DTC and INX-9 signal persisting (see white DTC outline in *Figure 1D*). We have since identified additional endogenous fluorescent-protein-tagged alleles that show expression in the gonadal sheath cells and localize in or near the cell membrane. One of these, *ina-1(qy23[ina-1::mNeonGreen])* (*Figure 2C*), was briefly reported in *Gordon et al., 2020*. We found another that marks the sheath, *cam-1(cp243[cam-1::mNeonGreen])* (*Heppert et al., 2018*; *Figure 2D*). For both tagged innexins as well as *ina-1::mNG* and *cam-1::mNG*, we find that the Sh1 cell has a distal boundary that either displays a measurable interface with the DTC or is so located as to be consistent with such a boundary (where the DTC is not marked by the endogenous protein). The position of this boundary (<40 μm, or ~8 germ cell diameters) coincides with the domain in which germ cells leave the stem cell niche (*Lee et al., 2019*) (purple shading in *Figure 2E*). We have not yet found a counterexample of an endogenously tagged, membrane-associated protein in Sh1 that demarcates an apparent Sh1 cell boundary at a great distance from the distal end of the gonad in young adults.

## Overexpressed transgenic markers vary in distal position and expression levels

Three integrated array transgene markers that drive overexpression of fluorescent proteins in the sheath were also analyzed. The first is a *lim-7* promoter-driven cytoplasmic GFP that was used to label the Sh1 cell in a foundational study of the *C. elegans* hermaphrodite gonad, *tnIs6[lim-7::GFP]* (*Hall et al., 1999*; *Figure 2F*). The second is a *lim-7* promoter-driven functional cell death receptor tagged with GFP, *bcIs39[lim-7p::ced-1::GFP]* (*Zhou et al., 2001*), which is the basis of a recent study that reports a more proximal boundary of Sh1 (*Figure 2G*, strain DG5020; *Tolkin et al., 2022*). The third is a *lim-7* promoter-driven membrane-localized GFP made by us to mark the sheath cell membrane without tagging an endogenous protein, *rlmIs5[lim-7p::GFP::CAAX]* (*Figure 2H*). The ranges of the distal edge of GFP localization for all three strains overlap with what we observed for the four endogenously tagged proteins, but are far more variable, as overexpressed transgenes are known to be (*Evans, 2006*; *Figure 2E–H*, and *Figure 2—figure supplement 1*). Patterns are more similar at earlier developmental stages (*Figure 2—figure supplement 2*).

To untangle this variance, we examined individual worms for evidence of a DTC-Sh1 interface. About half of the scoreable *lim-7p::ced-1::GFP* gonads (strain DG5020) show a DTC-Sh1 interface, and half show a bare region (*Figure 3A*). We further broke down this dataset by fluorescence intensity of distal CED-1::GFP signal. Strikingly, among animals under a threshold of expression intensity of ~400 AU (less than 1/3 as bright as the brightest GFP samples), the incidence of a DTC-Sh1 interface was 100% (10/10, as opposed to 15/30 for the whole dataset, *Figure 3A*). On the other extreme, gonads with stronger CED-1::GFP signal were more likely to have a farther proximal boundary of CED-1::GFP localization. In samples for which CED-1::GFP signal terminates at a great distance from the distal end of the gonad, there are two possible explanations. Either in those animals, the Sh1 position is farther proximal than in animals with other markers, or else CED-1::GFP fails to localize to the edge of the Sh1 cell pair.

## Expression differences between Sh1 cells in a pair can conceal distal extent of the sheath

We observed a pattern in a subset of gonads where the two Sh1 cells of a pair had dramatically different levels of CED-1::GFP signal, and these cells had different terminal positions on the distal-proximal axis (*Figure 3B and B'*). Exposure time and excitation laser power during image acquisition and subsequent scaling of the resulting image determine whether or not the signal in the lowly expressing cell is readily apparent (*Figure 3B vs B'*). In some cases, the brightness of the other Sh1 cell and the nearby proximal gonad makes the dimmer Sh1 cell nearly impossible to detect. Variable expression levels and even complete silencing of *C. elegans* transgenes are well-known phenomena (*Evans, 2006*). It was not known, however, that the two Sh1 cells of a pair could assume such different

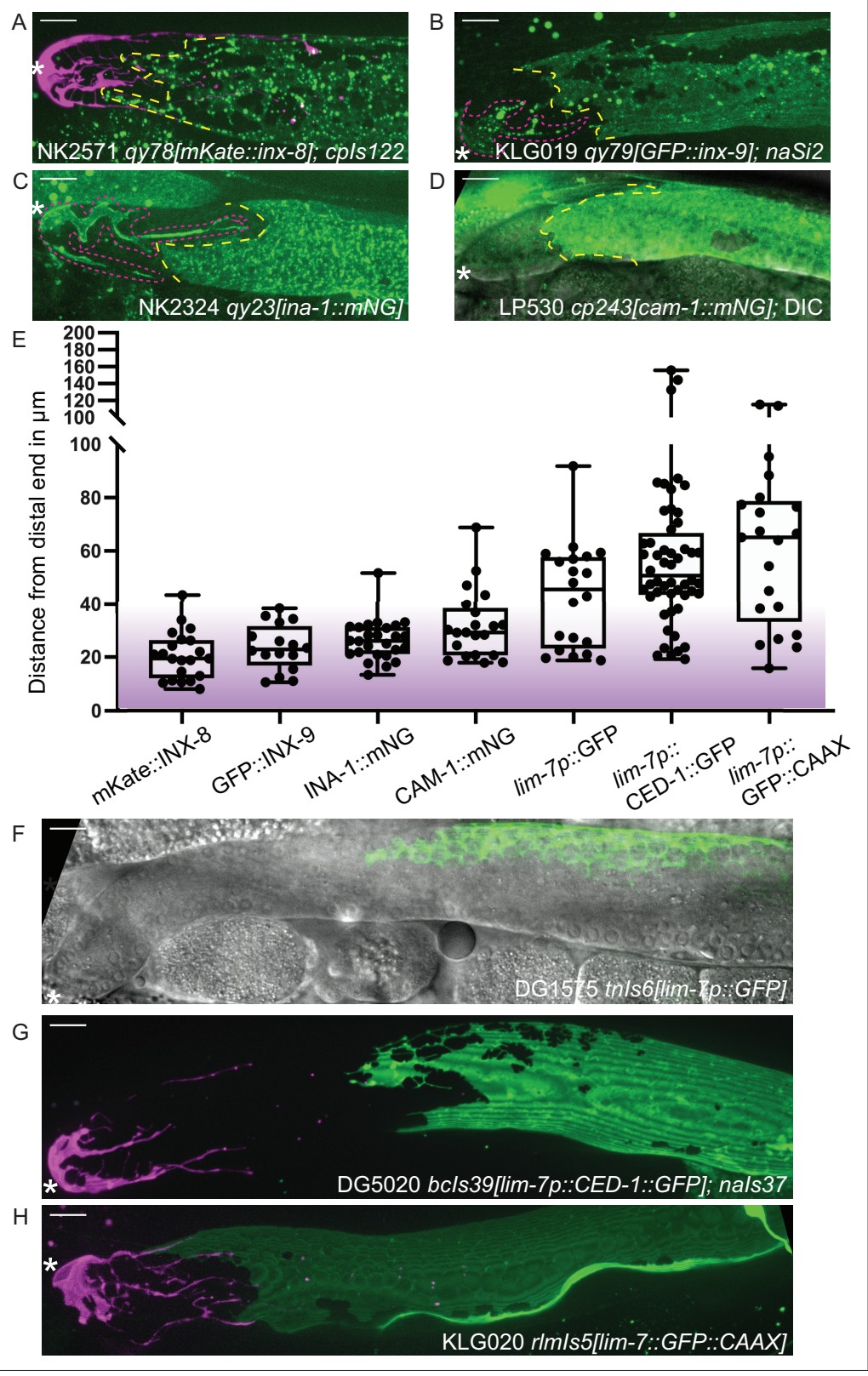

**Figure 2.** Sheath-expressed fluorescent proteins show consistency among endogenously tagged membrane proteins and greater variability in overexpressed transgenes. (**A**) NK2571 *qy78[mKate::inx-8]; cpIs122[lag-2p::mNeonGreen:: PLC^5PH]* N=21. (**B**) KLG019 *qy79[GFP::inx9];naSi2* (channel not shown) N=16. (**C**) NK 2324 *qy23[ina-1::mNG]* N=26. (**D**) LP530 *cp243[cam-1::mNG]* N=21. (**E**) Box plots of Sh1 positions for all strains listed

*Figure 2 continued on next page*

*Figure 2 continued*

above and below, with fluorescent protein listed on the graph, including transgenes. (**F**) DG1575 *tnIs6[lim-7p::GFP]* N=20. (**G**) Strain DG5020 *bcIs39[lim-7p::CED-1::GFP]; naIs37[lag-2p::mCherry-PH]* N=52 (note that mean and range agree with those reported in *Tolkin et al., 2022*). (**H**) KLG020 *rlmIs5[lim-7p::GFP::CAAX];cpIs122* N=21. Purple gradient marks approximate extent of stem cell zone (*Lee et al., 2019*; *Shin et al., 2017*). See *Figure 2— figure supplement 1* for images of minimum and maximum observed distances for all markers. *Figure 2—figure supplement 2* shows comparisons across development of NK2571 and DG5020. All scale bars 10 μm.

The online version of this article includes the following source data and figure supplement(s) for figure 2:

**Source data 1.** Source data used to generate plots of distal sheath measurements for strains shown in *Figure 2*.

**Figure supplement 1.** Endogenously tagged fluorescent proteins in the Sh1 membrane are less variable than overexpressed integrated transgenes.

**Figure supplement 2.** Differences between *qy78(mKate::inx-8)* and *bcIs39(lim-7p::ced-1::GFP)* expression in the sheath appear at the L4-young adult transition.

configurations over the distal germline (*Figure 3C*, and see *Figure 2H* for the same pattern in the *lim-7p::GFP::CAAX* transgenic strain).

The Sh1 positions become even more clear when *lim-7p::ced-1::GFP* is coexpressed with the mKate-tagged innexin *inx-8(qy78)* in strain DG5131 (*Figure 3D and E*). These markers colocalize in a substantial fraction of animals, as has been reported recently (*Tolkin et al., 2022*, see Figure 2— figure supplements 1 and 2 therein). In the animals that have a discrepancy between GFP and mKate localization in Sh1, the difference in expression reveals an unexpected cell boundary between the two Sh1 cells. We imaged 19 gonads from the coexpressing strain DG5131 through their full thickness. Of those, 4/19 had severe gonad morphology defects (see next section). Of the 15 morphologically normal gonads, 6/15 had discrepancies in CED-1::GFP and mKate::INX-8 signal. In 3/6 such cases, one Sh1 cell makes up the entire DTC-Sh1 interface, with the other terminating at a greater distance from the distal end. In the other 3/6 of cases, both Sh1 cells border the DTC. Fluorescence signal from mKate::INX-8 alone does not allow these cell borders to be detected because that marker is more consistently expressed across the Sh1 cells (*Figure 3F*).

The variability of the *lim-7p::ced-1::GFP* transgene allowed us to perform something like a mosaic analysis when the two Sh1 cells have very different expression levels but the dimmer cell is still visible (N=31/53 morphologically normal DG5020 gonads imaged to full depth, *Figure 3—figure supplement 1A-D*). Where the borders of the two Sh1 cells can be distinguished, one cell extends at least 20 μm farther distal than the other in 23/31 cases; five additional gonads have expression in only one Sh1 cell that terminates at a great distance (>70 μm) from the distal end. The edges of dimly expressing Sh1 cells can be difficult to resolve. A similar phenomenon was observed when the cytoplasmic GFP of *tnIs6[lim-7p::GFP]* was coexpressed with *qy78[mKate::inx-8]* (*Gordon et al., 2020*; Figure 1—figure supplement 1 therein). Of note, the N-terminal mKate::INX-8 and GFP::INX-9 tags are most likely extracellular based on the innexin-6 channel structure determined by cryo-EM (*Oshima et al., 2016*), so there is reason to suspect their localization at the cell membrane will be regulated differently than that of intracellular GFPs.

Additionally, we noticed that in DG5131 gonads where the two Sh1 cells have very different CED-1::GFP expression levels, sometimes mKate::INX-8 is missing from the membrane in Sh1 cells with strong CED-1::GFP signal (*Figure 3—figure supplement 1E and F*). Subtracting background, we find that there is a 50% reduction in tagged INX-8 in such membrane regions. Since mKate::INX-8 is a genomically encoded, functional protein, such disruption likely impacts endogenous protein function. This observation hints at a synthetic interaction between the two fluorescent sheath membrane proteins.

## Overexpression of CED-1::GFP transgene is correlated with gonad abnormalities

We therefore asked whether there was further evidence of a synthetic interaction between *lim-7p::ced-1::GFP* and *inx-8(qy78)*. First, we found evidence that suggests that *lim-7p::ced-1::GFP* is damaging to the animals with or without *qy78*. In the strain that expresses *lim-7p::ced-1::GFP* and not *qy78* (strain DG5020), roughly 20% of the animals had profound gonad migration defects in one

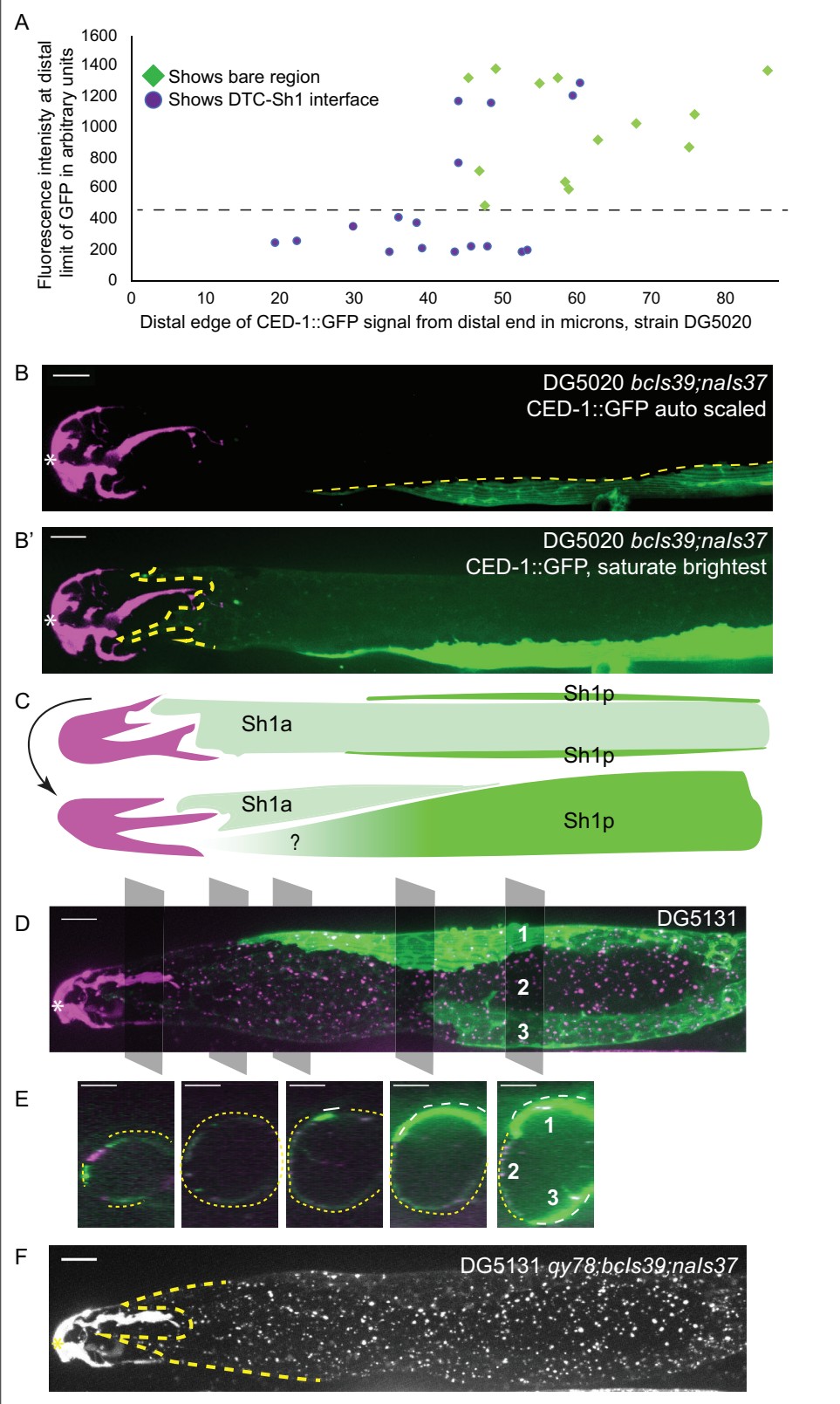

**Figure 3.** *lim-7p*::CED-1::GFP has variable expression intensity that conceals distal position of Sh1. (**A**) Plot of distal position vs. fluorescence intensity in arbitrary units of CED-1::GFP at the distal limit of its domain in N=30 DG5020 *bcIs39[lim-7p::CED-1::GFP]; naIs37[lag-2p::mCherry-PH]* animals. Dashed black line: all of the lowly expressing gonads (under ~400 AU, or <30% maximum brightness of brightest sample) have a distal tip cell (DTC)-

*Figure 3 continued on next page*

*Figure 3 continued*

Sh1 interface detected. (**B**) DG5020 sample in which disparate expression levels in the two Sh1 cells of a single gonad arm obscure detection of the DTC-Sh1 interface. The GFP channel is scaled automatically in B; B' is scaled to saturate the brightest pixels and reveal the dim second Sh1 cell. Dashed yellow link marks the edge of the bright Sh1 cell. (**C**) Schematic showing Sh1 pair configuration over distal germline, with the distal extent of Sh1p uncertain in superficial projection. The two Sh1 cells of a pair descend from the anterior and posterior daughters of Z1 and Z4, so the two Sh1 cells are here labeled Sh1a and Sh1p (arbitrarily). Top, superficial view. Bottom, side view. (**D**) DG5131 *qy78[mKate::inx-8]; bcIs39[lim-7p::CED-1::GFP]; naIs37[lag-2p::mCherry-PH]* sample in which one Sh1 cell contacts the DTC around the circumference of the germline and the other Sh1 cell lies at some distance from the distal end. Gray boxes and numbers mark planes and landmarks shown in (**E**). (**E**) Five cross sections through gonad in (**E**) made by projecting through two 1 µm re-slices at the positions shown by gray boxes in (**D**). Same analysis for DG5020 shown in *Figure 3—figure supplement 1*. (**F**) Same worm as in (**D,E**); signal from endogenously tagged allele *qy78[mKate::inx-8]* more uniformly labels the Sh1 cells, obscuring their individual shapes. All scale bars 10 µm.

The online version of this article includes the following source data and figure supplement(s) for figure 3:

**Source data 1.** Source data used to generate plots of distal sheath position and fluorescence intensity measurements for samples shown in *Figure 3A* and *Figure 4D*.

**Figure supplement 1.** The Sh1 cells of a pair can take two distinct configurations over the distal germline.

gonad arm (*Figure 4A and C*). We also observe such defects in the DG5131 strain that combines *qy78[mKate::inx-8]* with the *lim-7p::ced-1::GFP* transgene (*Figure 4B*, 4/19 or 21% of animals), so we cannot attribute this defect to a spontaneous mutation arising in a single population in transit. We have not observed such morphological defects in the original strain bearing *qy78*, nor in any other strain we have studied. The *lim-7p::ced-1::GFP* transgene seems to cause incompletely penetrant gonad morphology defects.

Whether or not overexpressed CED-1::GFP also disrupts the localization of untagged innexin proteins or other endogenous sheath membrane proteins as it does the tagged mKate::INX-8, and whether such disruption explains the gonad migration defects we observe for this allele, we currently cannot say. In many of these *qy78; lim-7p::ced-1::GFP* coexpressing animals (strain DG5131), the intensity of CED-1::GFP is notably low (*Figure 4D*). Lower expression levels of the CED-1::GFP fusion protein, with or without *qy78* in the background, appear more likely to reveal the distal Sh1 cell (*Figure 3A* and *Figure 4D*). This could either be because the absence of competing bright signal makes it easier to detect dimly expressing distal Sh1, or because high levels of the transgene product are not tolerated in the distal Sh1 cell. The overexpression of the functional cell death receptor CED-1, and not just the overexpressed membrane-localized GFP, could contribute to the defects observed in this strain. We sometimes observe abnormal sheath membrane protrusions that may result from aberrant engulfment of distal germ cells by the sheath (*Figure 4E*).

The discrepancy in apparent Sh1 position when two Sh1 cells express different amounts of CED-1::GFP and when CED-1::GFP is coexpressed with mKate::INX-8 provides definitive evidence that CED-1::GFP sometimes fails to label the entire distal sheath (the same phenomenon is reported in Figure 2—Figure Supplement 3B in the recent study *Tolkin et al., 2022*). Furthermore, the defects caused in gonads overexpressing this functional cell death receptor suggests that its localization to the distal Sh1 membrane at high levels is not well tolerated. We therefore conclude that *lim-7p::ced-1::GFP* is an unacceptable marker of distal Sh1.

## Assessing sheath markers for evidence of gonad disruption—brood size

Just because *lim-7p::ced-1::GFP* is a poor marker of the distal sheath does not, however, relieve concerns that the endogenously tagged innexins mKate::INX-8 and GFP::INX-9 are altering the gonad. A control for tagged innexin function was originally carried out (*Gordon et al., 2020*). Briefly, a careful genetic analysis (*Starich et al., 2014*) reported that the single mutant *inx-9(ok1502)* is fertile, but the *inx-8(tn1474); inx-9(ok1502)* double mutant is sterile. Therefore, attempts to use CRISPR/Cas9 to introduce a fluorescent tag in the *inx-8* locus were first performed in the *inx-9(ok1502)* background, and only once a fertile edited strain was recovered was the same edit introduced into the otherwise wild-type genetic background. We conducted brood size assays for strains discussed in this study, including the

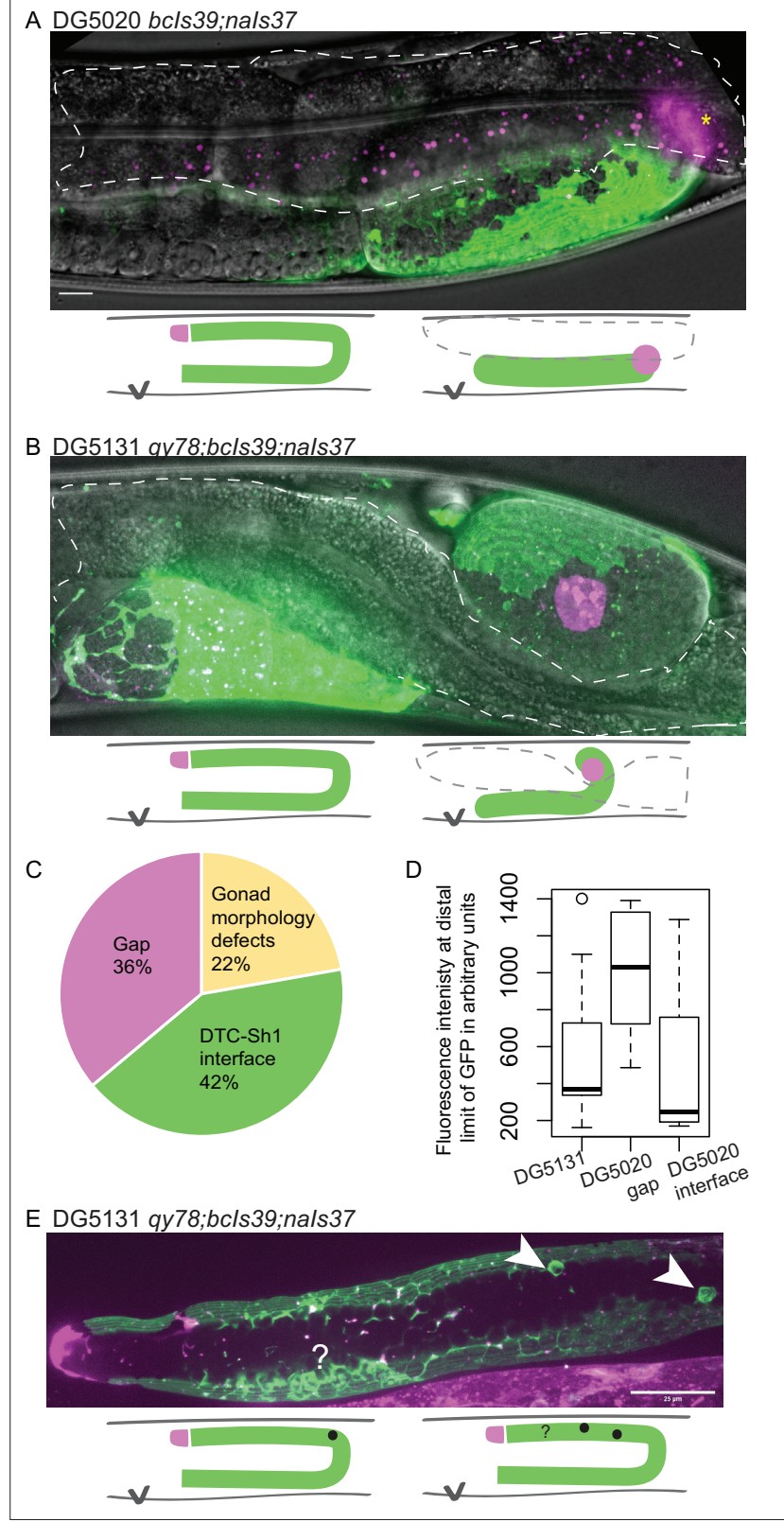

**Figure 4.** *lim-7p::ced-1::GFP* is correlated with gonad defects. (**A**) Example of gonad morphology defect in DG5020 *bcIs39[lim-7p::CED-1::GFP]; naIs37[lag-2p::mCherry-PH]* strain, in which the gonad failed to turn. Gut outlined in dashed shape; magenta puncta in that domain are autofluorescent gut granules. (**B**) Example of gonad morphology defect in DG5131 *qy78[mKate::inx-8]; bcIs39[lim-7p::CED-1::GFP]; naIs37[lag-2p::mCherry-PH]* strain,

*Figure 4 continued on next page*

*Figure 4 continued*

in which gonad turned once and arrested without elongating along the dorsal body wall. Schematics in A and B show wild-type gonad morphology with two turns and a distal tip cell (DTC) that arrives at the dorsal midbody, left, beside schematics of defective gonad migration shown in micrographs. (**C**) Relative proportions of phenotypes observed in DG5020 animals (N=72). (**D**) Boxplot comparing fluorescence intensity for coexpressing strain DG5131 in addition to data shown in *Figure 3* for DG5020. Fluorescence intensity of the *lim-7p::ced-1::GFP* transgene in this background is statistically indistinguishable from expression levels of this transgene in an otherwise wild-type background in the subset of samples that display a DTC-Sh1 interface, shown here segregated from samples from this strain that show a gap between the DTC and Sh1 cells. DG5131 N=17. DG5020 gap N=13. DG5020 interface N=17. A one-way ANOVA to determine the effect of category (genotype or presence of an interface) and fluorescence intensity was performed and was significant, $F_{2,44}=7.70$, p=0.001. Tukey's multiple comparison test finds that the mean fluorescence intensity of DG5020 gonads with a DTC-Sh1 interface differs from DG5020 gonads with a gap between Sh1 and the distal end (p=0.002) and does not differ from DG5131 worms (p=0.908). (**E**) Gonad from DG5131 strain with white arrowheads indicating aberrant engulfment of germ cells in the distal gonad. Closer to the distal end, a large mass of germ cells showing substantial localization of the CED-1::GFP protein may also reflect ectopic engulfment. Schematics show location of germ cell engulfment in wild-type gonads on the left and locations of the features marked in the micrograph in E on the right. Scale bars in A and B, 10 µm; scale bar in E, 25 µm.

The online version of this article includes the following source data for figure 4:

**Source data 1.** Classifications of 72 gonads from strain DG5020 that display a defect, a gap, or an interface used to generate pie chart in *Figure 4C*.

DG5131 strain containing both *lim-7p::ced-1::GFP* and the tagged innexin *qy78[mKate::inx-8]* that was imaged and analyzed by *Tolkin et al., 2022*, but not assayed for brood size (*Table 1*).

We find reductions in brood size for all of the strains under investigation, including a reduced brood size and notable embryonic lethality in two strains (DG5020 and DG5131) carrying the *lim-7p::ced-1::GFP* transgene. Interestingly, despite being genetically redundant genes (*Starich et al., 2014*) tagged in highly similar ways, and having similar live brood sizes, our endogenously tagged *inx-8(qy78) and inx-9(qy79)* strains had dramatically different degrees of embryonic lethality, with *qy79* producing over 150 unhatched eggs per worm. All of the fluorescently marked strains have mildly to moderately reduced brood sizes. On the basis of brood size alone, there is not a strong reason to prefer one of these markers over another.

**Table 1.** Brood size assays.

| Strain name | Full genotype | Live brood[*] | Reduction vs. wt % | Unhatched eggs | Embryonic lethality % |
|---|---|---|---|---|---|
| N2 | Wild type | 295±39 (n=57) | NA | NA | NA[†] |
| KLG019 | qy79[GFP::inx-9];nasi2[‡, §] | 226±22 (n=13) | 23% | 165±58 | 41 ± 8% |
| NK2571[¶] | *qy78[mKate::inx-8];cpIs122*§ | 220±41 (n=15) | 25% | 20±14 | 9 ± 5% |
| DG5020[¶] | *bcIs39[lim-7p::ced-1::GFP];naIs37*§ | 202±29 (n=12) | 32% | 62±47 | 20 ± 14% |
| DG5131 | qy78[mKate::inx-8];bcIs39[lim-7p::ced-1::GFP];naIs37§ | 187±45 (n=14) | 37% | 40±25 | 18 ± 11% |
| LP530 | *cp243[cam-1::mNG]* | 260±31 (n=10) | 12% | NA | NA |
| NK2324 | *qy23[ina-1::mNG]* | 237±37 (n=8) | 20% | NA | NA |

[*]Viable offspring that hatch from a single parent.

[†]N2 numbers come from multiple trials, not all of which were scored for embryonic lethality, including the trial in which *ina-1(qy23)* and *cam-1(cp243)* were counted.

[‡]qy79[GFP::inx-9] allele in strains NK2572 and NK2573 from *Gordon et al., 2020*, with germ cell nuclear marker *naSi2*; this combination of alleles was used in the cross to *glp-1(bn18)* in *Figure 1D*.

[§]Full transgene descriptions in Methods for germ cell (*naSi2*) and DTC (*cpIs122, naIs37*) markers.

[¶]See *Appendix 1—table 1* for replicates and statistical analysis of NK2571 and DG5020.

## Assessing sheath markers for evidence of gonad disruption— proliferative zone

Because brood size is an emergent property of many gonad, germline, embryonic, and systemic processes (including gonadogenesis, stem cell maintenance, regulation of meiosis, spermatogenesis, oogenesis, metabolism, ovulation, and embryogenesis), defects in brood size are not a direct proxy for dysregulation of the germline proliferative zone. We therefore turned our attention back to the distal gonad and asked whether the strains with fluorescent sheath markers have abnormalities in several metrics (*Figure 5A*). The length of the proliferative zone differs among strains (as measured by DAPI staining of germ cell nuclei to detect and measure the length of the germline distal to crescent-shaped nuclei of meiosis I, *Hubbard, 2007*; *Figure 5A, C and D*). The NK2571 strain with the tagged innexin *inx-8(qy78)* and DTC marker has a normally patterned distal germline (average proliferative zone length of 106 μm, or ~26 germ cell diameters) that is indistinguishable from wild-type N2 (average of 109 μm or ~27 germ cell diameters, *Figure 5C, C'*, *and D*). Excluding worms with gross morphology defects, the DG5020 strain bearing a DTC marker and *lim-7p::ced-1::GFP* has a measurably shorter distal germline (average of 91 μm, or ~23 germ cell diameters, *Figure 5C''*, *and D*). In the DG5131 strain that combines these alleles, the distal germline is notably shortened (average of 79 μm or ~20 germ cell diameters, *Figure 5C'''*, *and D*). This is comparable to the defect caused by the *glp-1(bn18)* allele at the permissive temperature shown in *Figure 1E*. Abnormal distal gonad patterning provides further evidence that a synthetic interaction between the *lim-7p::ced-1::GFP* transgene and the *qy78* allele—not the *qy78* allele alone—is responsible for the shorter proliferative zone observed for strain DG5131 (in agreement with Figure 4 from *Tolkin et al., 2022*).

We also counted the number of mitotic figures made by dividing cells in metaphase and anaphase in these strains (*Figure 5B*) and the total length of the gonad from vulva to tip (*Figure 5E*). Wild-type N2 had an average of 4.6 dividing cells per gonad; NK2571 had an average of 5.1 dividing cells per gonad (these two were not significantly different); DG5020 had an average of 3.4 dividing cells per gonad; DG5131 had an average of 3.2 dividing cells per gonad (these last two strains were significantly different from wild type, see *Figure 5B* and legend). Gonad lengths were not significantly different between N2 (average length of 670 μm), NK2571 (639 μm), or DG5020 (631 μm) but were significantly shorter in DG5131 (575 μm).

In the end, we find that only the strain combining *inx-8(qy78)* and *lim-7p::ced-1::GFP* has a dramatically smaller gonad that differs from the wild type in three key measures. Expression of the *qy78* allele alone with a DTC marker, on the other hand, does not cause any of these quantitative gonad phenotypes. The moderate brood size defects shown by all strains could be caused by numerous processes outside of stem cell regulation. For example, we find the hypothesis of *Tolkin et al., 2022*, based on the findings of *Starich et al., 2020* and *Starich et al., 2014*, that a major role of *inx-8/9* is in the proximal gonad regulating the provisioning of oocytes with essential metabolites, to be compelling. This hypothesis also has support from the large number of unhatched eggs observed for *inx-9(qy79[GFP::inx-9])*. Thus, we conclude with the observation that endogenous, fluorescently tagged sheath membrane proteins consistently mark both of the distal Sh1 cells without measurably impairing distal gonad function and should be the reagents of choice for live imaging in this cell type. They also consistently report a distal Sh1 position adjacent to the stem cell zone, as we previously found (*Gordon et al., 2020*).

## Discussion

We discovered that the distal position of Sh1 is much closer to the distal end of the young adult hermaphrodite gonad than than was previously observed, where it forms an interface with the DTC's proximal projections and overlaps substantially with the proliferative zone of the germline where mitotic cell divisions occur (*Gordon et al., 2020*). Importantly, that study did not claim that Sh1 is necessary or sufficient for what we term 'niche exit'; we simply observed that Sh1 associates with germ cells as they exit the niche by division. We have now confirmed this finding with functional manipulations of germ cell cycling and cell fate. We observed a distal Sh1 position in other strains with endogenously tagged sheath cell membrane proteins that act in molecular pathways outside of gap junctional coupling, and in a substantial fraction of traditional transgenic animals expressing *lim-7* promoter-driven CED-1::GFP, GFP::CAAX and cytoplasmic GFP (though these strains have high

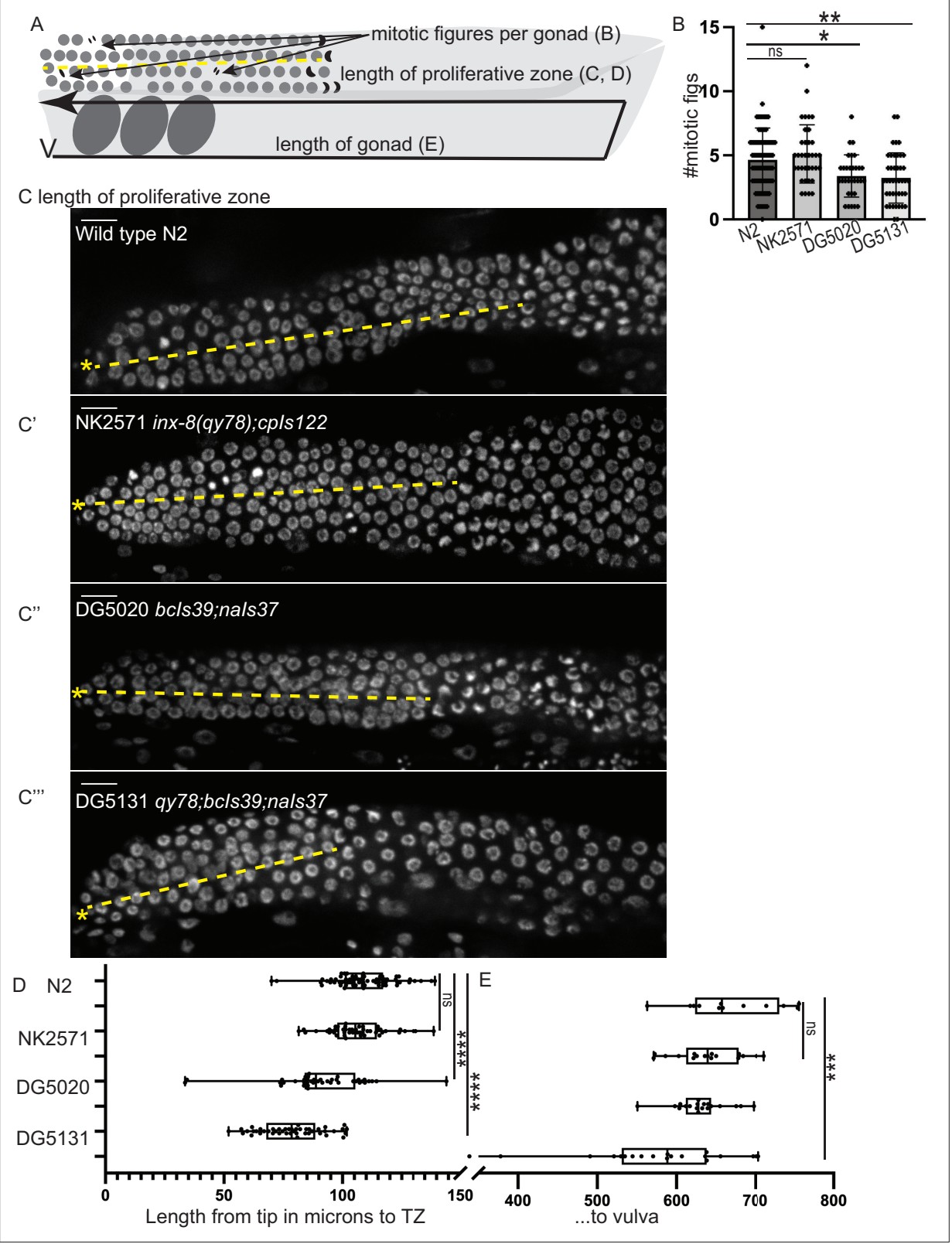

**Figure 5.** A synthetic interaction between *lim-7p::ced-1::GFP* and the tagged innexin *qy78* shortens the proliferative zone. (**A**) Illustration of measurements made for *Figure 5*. (**B**) Number of mitotic figures observed in DAPI stained animals of the four strains. Numbers of dividing cells and gonads examined are as follows: wild-type N2 (N=311 dividing cells/67 gonads), the NK2571 strain with the tagged innexin *qy78* (N=184 dividing cells/36 gonads), the DG5020 strain with *lim-7p::ced-1::gfp* (N=105 dividing cells/31 gonads), the DG5131 strain combining these sheath markers

*Figure 5 continued on next page*

*Figure 5 continued*

(N=136 dividing cells/42 gonads). A one-way ANOVA to determine the effect of genotype on number of mitotic figures was significant F3, 172=7.081, p=0.0002. Tukey's multiple comparison test revealed that NK2571 did not differ from wild type (mean difference –0.47 cells per gonad, 95% CI –1.65 to 0.71, p=0.7291), DG5020 differed from wild type by 1.26 germ cells per gonad, 95% CI 0.02–2.50, p=0.0453, and DG5131 differed from wild type (mean difference of 1.40 cells per gonad, 95% 0.28–2.52, p=0.0075). (**C-C'''')** DAPI stained distal gonads for measurement of proliferative zone for the four strains. Asterisk marks tip of gonad, dashed line marks example of lengths measured. (**C**) Wild-type N2 (N=68), (**C'**) the NK2571 strain with the tagged innexin *qy78* (N=49), (**C''**) the DG5020 strain with *lim-7p::ced-1::gfp* (N=40), (**C'''**) and the DG5131 strain combining these sheath markers (N=45). Asterisk marks tip of gonad. (**D**) Plots of proliferative zone length (left) and whole gonad length (right) for the four strains. A one-way ANOVA to determine the effect of genotype on length of proliferative zone was significant F3,198=49.15, p<0.0001. Tukey's multiple comparison test revealed that NK2571 did not differ from wild type (mean difference 2.69 μm, 95% CI –4.283 to 9.663 μm, p=0.750), DG5020 differed from wild type by ~2–5 germ cell diameters (mean difference of 17.94 μm, 95% CI 10.53 to 25.36 μm, p<0.0001), and DG5131 dramatically differed from wild type (mean difference of 30.36 μm, 95% CI 23.21 to 37.51 μm, p<0.0001). The proliferative zone length of DG5131 was also significantly different from both of its parent strains (NK2571 vs. DG5131 mean difference of 27.67 μm, 95% CI 19.99 to 35.35 μm, p<0.0001; DG5020 vs. DG5131 mean difference of 12.42 μm, 95% CI 4.331 to 20.50 μm, p=0.0006). (**E**) Plots of length of entire gonad from tip to vulva. N2 (N=12), NK2571 (N=17), DG5020 (N=19), DG5131 (N=20). A one-way ANOVA to determine the effect of genotype on gonad length was significant F3,64=6.27, p=0.0009. Tukey's multiple comparison test revealed that NK2571 did not differ from wild type (mean difference 31.16 μm, 95% CI –32.99 to 95.32 μm, p=0.578), DG5020 also did not differ from wild type (mean difference of 39.42 μm, 95% CI –23.32 to 102.2 μm, p=0.3546), and DG5131 did differ from wild type (mean difference of 95.15 μm, 95% CI 33.02 to 157.3 μm, p=0.0008). All scale bars 10 μm.

The online version of this article includes the following source data for figure 5:

**Source data 1.** Measurements used to generate plots of proliferative zone length, gonad length, and number of mitotic figures for **Figure 5**.

variability in fluorescence intensity and localization). Therefore, we consider the results presented here to be confirmatory of the foundational finding of *Gordon et al., 2020*, which is that almost all mitotic germ cells in the adult hermaphrodite contact the DTC or Sh1, with a noteworthy population in contact with both. Other recent work provides further evidence of a role for sheath cell contact in promoting adult germ cell proliferation, specifically through modulation of Notch receptor *glp-1* expression (*Gopal et al., 2020*). We focus especially on young adults in these studies (less than 24 hr post mid-L4, see Methods). An important caveat to the work is that the gonad is dynamic, and cell shapes and positions change over time. Indeed, dynamic processes could lead to the surprising difference in position often seen between the two Sh1 cells in a single gonad arm, if one Sh1 cell grows more actively over germ cells as they leave the niche. The high variability of expression levels of an overexpressed *lim-7p::ced-1::GFP* transgene has allowed for this surprising discovery, though that variability makes it a poor marker of the absolute position of the Sh1 cells, it sometimes causes gonad defects, and it interacts synthetically with *qy78* to cause germline defects.

This work was inspired by a recent preprint (*Tolkin et al., 2021*; *biorxiv*, version 1). This preprint initially reported a severe brood size and embryonic lethality defect in strains bearing the *qy78* allele (just over 100 offspring per animal, with day 1 embryonic lethality of nearly 90%) and hypothesized that abnormal innexin function caused by endogenously tagging INX-8 with mKate could be responsible for both the fertility defect and the novel finding of a distal position of Sh1 abutting the stem cell zone. This indeed would be a serious concern, and we are grateful that other researchers in the community are vigilant and interested in the strains we generated.

*Tolkin et al., 2022* (revised) downscales concerns about brood size (over 200 offspring per animal) and embryonic lethality (under 10% total) but still proposes that the distal Sh1 position revealed by *qy78* is an artifact of protein tagging. The study uses an overexpressed, GFP-tagged, functional cell death receptor protein as the preferred marker of the sheath (the *bcls39* allele encoding *lim-7p::ced-1::GFP*), but does not justify why such an element should be assumed to be a robust and non-phenotypic marker of the distal sheath. This genetic element—which we demonstrated to interact synthetically with *qy78* to cause gonad and germline defects (*Figure 5*)—is present in the genetic background of the worms analyzed in the vast majority of the experiments reported in the four data figures of *Tolkin et al., 2022*: Figure 1 (all, no true wild-type analyzed), Figure 2 (half of panels D–F with another marker in the other half of those panels, see next), Figure 3 (all, no wild-type analyzed), Figure 4 (all, no wild-type analyzed). In all of these experiments, we conclude the synthetic interaction is driving the phenotypes attributed to *qy78*, including the dosage-dependent and deletion-mediated suppression of the phenotypes attributed to *qy78* shown in *Tolkin et al., 2022* (Figure 2—figure supplement 2).

In the rest of Figure 2D-F, Tolkin et al. use a *fasn-1::GFP* sheath marker coexpressed with *qy78*. Only six samples were observed, and only 2/6 had the 'Class 3' phenotype with a dramatically distal Sh1 border. This differs notably from what is observed for the innexin-defective *inx-14(ag17)* hypomorphic allele shown in Figure 1—figure supplement 1C for which 21/21 *fasn-1::GFP* coexpressing samples had 'Class 3' gonads. We do not find this experiment to be decisive because of low sample size for this genotype and results in Figure 2F that appear to fall within the distribution of the control.

Next, we consider the other experiments that do not include the *bcIs39* marker. In Figure 2B–C and Figure 2—figure supplements 1 and 2, Tolkin et al. use an anti-INX-8 antibody in an immunofluorescence experiment that reports a more proximal Sh1 boundary in the wild type than in worms carrying the *qy78* allele. We note the conspicuous absence of innexin detected in the DTC, where it had been reported previously (see the wild-type gonad image in both Figure 3 of *Starich et al., 2014*, and Figure 2A of *Starich et al., 2020*). We also note that the anti-INX-8 sheath localization is like a honeycomb, not sparse and punctate. *Starich et al., 2020* present evidence that a honeycomb localization pattern for innexins indicates that gap junctions are not forming properly, so this pattern in the wild-type sample is unexpected. Different imaging modalities often yield different patterns, and the previously published control image was made on a compound microscope while the image in *Tolkin et al., 2022* (Figure 2B) was made with a confocal microscope. However, it does not seem likely that a switch in imaging approaches would detect more abundant signal in one region (Sh1) and yet lose signal in another (the DTC) in the same field of view. Setting aside questions of reproducibility of this antibody staining experiment, this type of data could shed light on the position of the sheath in a genetically wild-type animal. Of note, Figure 3C of *Starich et al., 2014* shows antibody staining of gap junctions forming in what appears to be the Sh1 region at a distance of ~10 germ cell diameters from the distal end.

Transmission electron microscopy was also included in the revised version of *Tolkin et al., 2022*. The 3D reconstruction in Figure 1—figure supplement 2 shows a sheath that has a honeycomb pattern, presumably due to the thinness of Sh1 as it overlies germ cell bodies (annotation was based on the presence of mitochondria), so we know this technique under-annotates the thinnest regions of the Sh1 cell. It also appears to show what *Tolkin et al., 2022*, would call a 'Class 2' gonad—the DTC and Sh1 cells terminate within ~2 germ cell diameters of one another. It is not representative of a 'Class 1' gonad with a large gap (that 'exceeds 25 µm'). According to the scaling in Figure 1—figure supplement 2, the distal Sh1 annotations fall ~40 µm from the distal end. Taken at face value, this TEM data is equivocal on the question of whether there is a 'bare region' between the DTC and Sh1.

When we inspect the TEM stack in Video 4, we observe unannotated structures surrounding the germ cells. They have low complexity, making us wonder if they are an artifact of tissue shrinkage during fixation. Some appear to be contiguous with the annotated somatic cells (DTC and Sh1). Perhaps a more generous annotation would reveal more of the thin somatic cell structures that we know from live imaging are there (minimally, continuous Sh1 cover across germ cell bodies in the more proximal region). With that level of annotation, what distal Sh1 structures would appear? This is a very helpful type of data to include, and hopefully future studies of more TEM samples will help decide the issue.

Finally, each study furnishes additional strains with Sh1 fluorescence expression terminating in variously distal or more proximal domains: Tolkin et al. use *fasn-1::gfp* (endogenously tagged, cytoplasmic), *acy-4::gfp* (extrachromosomal, membrane localized), and *lim-7p::gfp* (integrated array, cytoplasmic); we report *ina-1::mNG* (endogenously tagged, membrane localized), *cam-1::mNG* (endogenously tagged, membrane localized), and *lim-7p::GFP::CAAX* (integrated array, membrane localized). The challenge with definitively proving a more proximal boundary of Sh1—after seeing images like those described in our *Figures 2H and 3* and its *Figure 3—figure supplement 1*, *Gordon et al., 2020* (*Figure 1—figure supplement 1C*) , and *Tolkin et al., 2022* (Figure 2—figure supplement 3) —is the challenge of proving a negative. When distal expression is not observed, how can one be certain that the whole sheath is labeled? Absence of evidence is not evidence of absence, and the aforementioned images make it clear that some fluorescent markers fail to capture distal Sh1 structures even when the structures are detectable by other means, and even when they brightly label more proximal cells.

Taken together, we suspect that we are seeing more or less the same things but describing them differently. *Tolkin et al., 2022* observe a DTC-Sh1 interface in many worms (~30%) expressing their

favored sheath marker (the highly variable *bcIs39*). These 'Class 2' gonads have the pattern that we first reported for endogenously tagged sheath proteins (see *Gordon et al., 2020*, *Figure 1B*), including *ina-1::mNG* (an integrin subunit) and *cam-1::mNG* (a Wnt pathway member) that are not obvious candidates to have dominant antimorphic effects on innexin signaling. *Tolkin et al., 2022* indeed identify several genetic backgrounds in which the spatial relationship between the DTC and Sh1 is perturbed, though how the mechanism acts through changes in innexin function, CED-1::GFP overexpression, or both, remains to be seen. We strongly agree that innexins in the gonadal sheath are important for gonad and germline development.

In physics, the observer effect states that it is impossible to observe a system without changing it. In biological imaging in *C. elegans*, this means that we can either observe wild-type animals that are dead, dissected and/or fixed and coated or stained, or we can observe genetically modified animals that are alive. Some fine, membranous cellular structures do not survive fixation (*Gerdes et al., 2013*; *Kornberg and Roy, 2014*). On the other hand, any genomic modification runs the risk of altering an animal's physiology.

We feel most confident examining endogenously tagged gene products in Sh1 for several reasons. First, proteins expressed at physiological levels are less likely to directly damage a cell vs. overexpressed fluorescent proteins (*Kintaka et al., 2016*). Second, the ability to cross-reference among strains with different tagged proteins that act in different molecular pathways allows us to use concordant results in reconstructing cell positions; any single marker may or may not localize to the region of interest, but concordant results among independent experiments help construct an accurate picture of the cell. One factor to consider, however, is that not every endogenously expressed protein is likely to localize evenly across all regions of a cell. We would expect in a large cell like Sh1 that interacts with germ cells in many stages of maturation that some cell-surface proteins would be regionalized. Along those lines, it seems possible that the Sh1 cells might have mechanisms to exclude the cell death receptor CED-1 from the cell membrane domain that contacts proliferating germ cells. The *bcIs39* transgene is typically used to study engulfment of apoptotic germ cell corpses at the bend of the gonad and rescues *ced-1* loss-of-function mutants for apoptotic germ cell corpse engulfment (*Zhou et al., 2001*). We find this marker to be unreliable in the distal region of the cell, and to cause gonad defects especially but not only when combined with endogenously tagged *inx-8(qy78)*. A recent study (*Tolkin et al., 2022*) uses this transgene in most of the backgrounds analyzed (sometimes detecting the CED-1::GFP by anti-GFP antibody staining, which appears to amplify the variability of the marker), so we find this problematic reagent to undermine that study's conclusions.

The need for caution when observing and interpreting endogenously tagged fluorescent proteins is noted. Several steps can and should be taken to increase confidence that a tagged protein is not causing cryptic or unwanted phenotypes. First, multiple edited lines should be recovered and outcrossed, thereby reducing the likelihood that a phenotype is caused by off-target Cas9 cutting creating lesions in any individual edited genome. Second, brood size should be estimated either by timed food depletion (less rigorous) or formal brood size assays (more rigorous). Third, edited lines should be examined for known phenotypes caused by loss of function of the targeted genes. This can be done, in order of least to most rigorous, by consulting the literature, by comparing to RNAi treatments or known mutants, and finally by introducing AID tags and using the degron strategy to deplete the gene product under the lab's exact experimental conditions of choice (*Zhang et al., 2015*), however this step will not work for extracellular tags (because extracellular AID tags are not accessible to TIR1 ubiquitin ligase). Finally, any 'markers' used should be assessed on their own for phenotypes. Even with these controls in place, synthetic interactions can emerge between 'markers' and alleles, including tagged proteins of interest. These interactions can themselves reveal biologically relevant phenomena, but only if they are recognized.

In the end, no transgenic or genome-edited strain is truly wild type, and it should be our expectation that such strains might be somewhat sensitized as a result. Indeed, the synthetic interaction we document between *lim-7p::ced-1::gfp* and *inx-8(qy78)* suggests that the *qy78* is sensitized for gonad defects caused by other genetic elements affecting the gonadal sheath. However, the perfect reagent does not exit. We can only look for congruent results among a set of independent reagents with non-overlapping weaknesses. Finally, we can formulate questions narrowly enough that, despite their shortcomings, our imperfect reagents are adequate to help answer them. In the future, new endogenously tagged alleles that are expressed in the sheath, single-copy, membrane-localized transgenes

that do not affect distal gonad patterning, and different imaging modalities like electron microscopy will shed more light on the complex relationship between the gonadal sheath and the germline. At the present time, however, we consider the existence of an interface between the DTC and Sh1 cells that coincides with the boundary of the distal-most stem-like germ cells to be supported by the preponderance of evidence.

## Methods

### Strains

In strain descriptions, we designate linkage to a promoter with a *p* following the gene name and designate promoter fusions and in-frame fusions with a double semicolon (::). Some integrated strains (*xxIs* designation) may still contain for example the *unc-119(ed4)* mutation and/or the *unc-119* rescue transgene in their genetic background, but these are not listed in the strain description for the sake of concision, nor are most transgene 3' UTR sequences.

### Staging of animals for comparisons among sheath markers

We focused on young adult animals around the time egg laying commences, as in *Gordon et al., 2020*. Mid L4 animals are picked from healthy, unstarved populations (which are maintained without starving for the duration of the experiment). These animals are kept at 20°C for 16–18 hr, until adulthood is reached and ovulation begins. We prefer not to age the animals much farther into adulthood for routine imaging (though we did this for the temperature shift experiments to follow previously published experimental regimes), as once a full row of embryos is present in the uterus, the distal gonads can become compressed or obscured by embryos. For strains in which a gonad migration defect is observed (DG5020, DG5131), picking animals in the L4 stage prevents bias for or against normal-looking adults (as the defects are profound enough to be visible on the dissecting scope in adults).

### Temperature-sensitive mutant analysis

Worms from the *emb-30(tn377)* mutant genotype were grown at the permissive temperature (16°C) for 24 hr past L4. Plates were shifted to the restrictive temperature (25°C) for 15 hr before DAPI staining, while permissive temperature controls were maintained at 16°C for 18 hr before staining (because development is proportionally slower at 16°C than at 25°C, permissive temperature controls were cultured longer). Two replicates of this experiment were performed with the results combined in *Figure 1E*. A starting point control (as in *Cinquin et al., 2010*) and a 21 hr control were also performed, with congruent results (*Figure 1—figure supplement 1*).

Worms from the *glp-1(bn18)* mutant genotype were grown at the permissive temperature of 16°C for 24 hr past L4. Plates were shifted to the restrictive temperature (25°C) for 6 hr (*Fox and Schedl, 2015*). Permissive temperature controls were maintained at 16°C for 6 hr. Worms were imaged live (see *Confocal imaging*, below).

### DAPI staining

DAPI staining was done by modifying standard protocols (*Francis and Nayack, 2000*), with the cold methanol fixation done for a shorter time (2.5 min) and the concentration of DAPI higher at 1 µg/ml in 0.01% Tween in PBS in the dark for 5 min, washed once with 0.1% Tween in PBS. Samples were briefly stored at 4°C in 75% glycerol and imaged directly in glycerol solution.

### Confocal imaging

All images were acquired on a Leica DMI8 with an xLIGHT V3 confocal spinning disk head (89 North) with a ×63 Plan-Apochromat (1.4 NA) objective and an ORCAFusion GenIII sCMOS camera (Hamamatsu Photonics) controlled by microManager (*Edelstein et al., 2010*). RFPs were excited with a 555 nm laser, GFPs were excited with a 488 nm laser, and DAPI was excited with a 405 nm laser. Worms were mounted on agar pads with 0.01 M sodium azide (live) or in 75% glycerol (DAPI stained).

### Fluorescence intensity of lim-7p::CED-1::GFP and mKate::INX-8

For quantitative comparisons of fluorescence intensity shown in *Figure 3* and *Figure 4*, gonads were imaged with uniform laser power and exposure times with 1 µm Z-steps. Images were opened in FIJI

(*Schindelin et al., 2012*) and z-projections were made through the depth of the superficial half of the gonad (not including signal from the deep Sh1 cell if it was present). Images without any detectable Sh1 expression were discarded (2/32 images from the analysis in *Figure 3A*). A line ~20 μm long parallel to long axis of the gonad, terminating near the distal boundary of GFP expression, and not crossing any gaps in Sh1 revealing background was drawn, and average fluorescence intensity was measured along its length in arbitrary units.

## Measurements of DTC and Sh1 positions

The distal tip of the gonad was identified in the fluorescence images if the DTC was marked or in a DIC image if the DTC was not marked in a given strain. The distance from the gonad tip to the longest DTC process (when marked), and from the gonad tip to the most distal extent of Sh1 was measured in FIJI (*Schindelin et al., 2012*). A DTC-Sh1 interface is detected by subtracting the first value from the second value—negative numbers reflect the amount of overlap of these cellular domains across the germline, positive numbers reflect a gap. This is a conservative estimate, as a gap of less than 1 germ cell diameter (~5 μm) would still allow germ cells to contact both the DTC and Sh1 at the same time. Min/max settings on the fluorescence images are adjusted to allow the faintest signal to be detected when measuring.

## Analysis of mosaic expression

The variability of the *lim-7p::ced-1::gfp* transgene allowed us to distinguish the two Sh1 cells in a pair, especially when coexpressed with *qy78[mKate::inx-8]*. For this experiment, we imaged animals through the full thickness of the distal gonad (40 μm instead of our usual 20 μm that captures just the superficial half of the gonad that can be imaged more clearly). Animals in which two distinct Sh1 cells had different levels of GFP signal were analyzed further for relative cell position. For DG5131, this was 6/19 samples. For DG5020, this was 31/53 samples.

## Brood size assays

DG5020 and DG5131 were shipped overnight on 9/23, passaged off the starved shipment plate onto fresh NGM+OP50 plates and maintained by passaging unstarved animals for three generations before beginning the brood size assay. For each strain, 10–15 L4 animals were singled onto NGM plates seeded with OP50 and kept in the same incubator, on the same shelf, at 20°C. The singled animals were passaged once per day on each of the following 5 days to fresh plates, with all plates maintained at 20°C. Two days after removing the parent, the plates with larval offspring were moved to 4°C for 20 min to cause worm motion to cease, and all larvae (and unhatched eggs when noted) were counted on a dissecting scope with a clicker by the same team of worm counters, with internal controls. Plates with unhatched eggs were examined and recounted 1 day later to see if any hatched. Offspring from parent worms that died or burrowed in the process were not counted. Total sample sizes and results reported in *Table 1*. Replicate brood sizes for DG5020 and NK2571 were performed by a neighboring lab (Dr R Dowen) with the strain names anonymized (*Appendix 1—table 1*).

## Distal germline patterning and mitotic figures

Measurements were made in FIJI from the distal end of the gonad to the transition zone, which is the distal-most row of germ cells with more than one crescent-shaped nucleus. Mitotic figures were counted manually as metaphase or anaphase DAPI bodies. Observations of 0 mitotic figures were counted in the analysis. For *Figure 1—figure supplement 1E*, measurements were made by manually counting cell diameters. In the distal-most region of the restrictive temperature samples, germ cell nuclei are abnormal, so absolute distances in microns were divided by the diameter of a normal-looking germ cell from the distal end to calculate germ cell diameters in this region.

## Gonad length measurements

Strains were synchronized by bleaching and L1 larvae transferred to OP50 seeded NGM plates. At 16°C for 48 hr. L4 worms were picked to fresh plates and cultured at 16°C for an additional 24 hr. These Day 1 adult worms were mounted on agar pads with 0.01 M sodium azide and imaged live. Images were analyzed in FIJI using the segmented line tool from vulva to distal gonad tip (usually in two tiled images to cover the whole gonad length).

## Statistical analyses

Tests, test statistics, and p values given for each analysis in the accompanying figure legends. One-way ANOVA followed by Tukey's multiple comparisons test were conducted in R (*R Development Core Team, 2020*) or Prism (GraphPad Prism version 9.20 (283) for macOS), GraphPad Software, San Diego, CA.

## Acknowledgements

We thank T Tolkin, A Mohammed, T Starich, Ken CQ Nguyen, David H Hall, T Schedl, JA Hubbard, and D Greenstein for sharing their manuscript and strains DG5020 (combining published alleles bcls39 and nals37) and DG5131 (combining published alleles qy78, bcls39, and nals37). We thank D Greenstein and the CGC for the temperature-sensitive mutant strains and B Goldstein and A Pani for LP530. We thank R Dowen and P Breen for anonymized brood size replication experiments. We are grateful for helpful conversations with D Sherwood and other colleagues. Funded by NIGMS Grant 1R35GM147704-01 to KLG.

---

## Additional information

### Funding

| Funder | Grant reference number | Author |
|---|---|---|
| National Institute of General Medical Sciences | 1R35GM147704-01 | Kacy Lynn Gordon |

The funders had no role in study design, data collection and interpretation, or the decision to submit the work for publication.

### Author contributions

Xin Li, Noor Singh, Data curation, Formal analysis, Investigation, Visualization, Writing - review and editing; Camille Miller, Data curation, Supervision, Investigation, Writing - review and editing; India Washington, Bintou Sosseh, Investigation, Writing - review and editing; Kacy Lynn Gordon, Conceptualization, Data curation, Formal analysis, Validation, Investigation, Visualization, Methodology, Writing - original draft, Project administration, Writing - review and editing

### Author ORCIDs

Kacy Lynn Gordon http://orcid.org/0000-0003-0967-4020

### Decision letter and Author response

Decision letter https://doi.org/10.7554/eLife.75497.sa1
Author response https://doi.org/10.7554/eLife.75497.sa2

---

## Additional files

### Supplementary files

• Transparent reporting form

### Data availability

Source data files contain the numerical data used to generate the figures.

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

# Appendix 1

## Appendix 1—key resources table

| Reagent type (species) or resource | Designation | Source or reference | Identifiers | Additional information |
|---|---|---|---|---|
| Gene (*Caenorhabditis elegans*) | *inx-8* | https://wormbase.org/ | Sequence CELE_ZK792.2 | Encodes gap junction hemichannel subunit |
| Gene (*Caenorhabditis elegans*) | *inx-9* | https://wormbase.org/ | Sequence CELE_ZK792.3 | Encodes gap junction hemichannel subunit |
| Gene (*Caenorhabditis elegans*) | *glp-1* | https://wormbase.org/ | Sequence CELE_F02A9.6 | Encodes Notch receptor |
| Gene (*Caenorhabditis elegans*) | *emb-30* | https://wormbase.org/ | Sequence CELE_F54C8.3 | Encodes putative member of APC |
| Gene (*Caenorhabditis elegans*) | *ina-1* | https://wormbase.org/ | Sequence CELE_F54G8.3 | Encodes worm alpha integrin ortholog |
| Gene (*Caenorhabditis elegans*) | *cam-1* | https://wormbase.org/ | Sequence CELE_C01G6.8 | Encodes worm Wnt receptor |
| Gene (*Caenorhabditis elegans*) | *ced-1* | https://wormbase.org/ | Sequence CELE_Y47H9C.4 | Encodes worm cell death receptor |
| Genetic reagent (*Caenorhabditis elegans*) | *inx-8(qy78(mKate::inx-8)) IV; cpIs122(lag-2p::mNeonGreen:: PLC^{dPH})* | **Gordon et al., 2020** | NK2571 | Can be obtained from K Gordon lab |
| Genetic reagent (*Caenorhabditis elegans*) | *inx-9(qy79(GFP::inx-9)) IV; naSi2(mex-5p::H2B::mCherry::nos-2 3'UTR) II* | *nasi2* transgene from **Roy et al., 2018**; *qy79* from **Gordon et al., 2020** | KLG019 | Can be obtained from K Gordon lab |
| Genetic reagent (*Caenorhabditis elegans*) | *rlmIs5[lim-7p::GFP::CAAX]* | This study | KLG020 | Can be obtained from K Gordon lab |
| Genetic reagent (*Caenorhabditis elegans*) | *qy78(mKate::inx-8) IV* | This study | KLG021 | ×2 outcross of NK2571 to N2 |
| Genetic reagent (*Caenorhabditis elegans*) | *inx-9(qy79(GFP::inx-9)) IV; naSi2(mex-5p::H2B::mCherry::nos-2 3'UTR) II; glp-1(bn18) III* | *glp-1(bn18)* from **Kodoyianni et al., 1992** doi: 10.1091/mbc.3.11.1199 | KLG022 | Mutant obtained from CGC, crossed to KLG019 |
| Genetic reagent (*Caenorhabditis elegans*) | *inx-8(qy78(mKate::inx-8)) IV; cpIs122(lag-2p::mNeonGreen:: PLC^{dPH}); emb-30(tn377) III* | *emb-30(tn377)* from **Cinquin et al., 2010** doi: 10.1073/pnas.0912704107 | KLG023 | Mutant obtained from CGC, crossed to NK2571 |
| Genetic reagent (*Caenorhabditis elegans*) | *cp243[cam-1::mNG]* | **Heppert et al., 2018** doi:10.1534/GENETICS.117.300487 | LP530 | Can be obtained from B Goldstein lab |
| Genetic reagent (*Caenorhabditis elegans*) | *qy23[ina-1::mNG]* | **Jayadev et al., 2019** doi: 10.1083/jcb.201903124 | NK2324 | Can be obtained from D Sherwood lab |

*Appendix 1 Continued on next page*

*Appendix 1 Continued*

| Reagent type (species) or resource | Designation | Source or reference | Identifiers | Additional information |
|---|---|---|---|---|
| Genetic reagent (*Caenorhabditis elegans*) | *tnIs6[lim-7p::GFP]* | *Hall et al., 1999* | DG1575 | Obtained from CGC |
| Genetic reagent (*Caenorhabditis elegans*) | *bcIs39[lim-7p::ced-1::GFP];naIs37* | *Tolkin et al., 2022*; *naIs37* originally from *Pekar et al., 2017* | DG5020 | See *Tolkin et al., 2022* |
| Genetic reagent (*Caenorhabditis elegans*) | *qy78[mKate::inx-8];bcIs39[lim-7p::ced-1::GFP];naIs37* | *Tolkin et al., 2022* | DG5131 | See *Tolkin et al., 2022* |
| Software, algorithm | µManager software v1.4.18 | (*Edelstein et al., 2010*) doi: 10.1002/0471142727.mb1420s92 | RRID:SCR_016865 | https://micro-manager.org/ |
| Software, algorithm | FIJI 2.0 | *Schindelin et al., 2012* doi: 10.1038/nmeth.2019 | RRID:SCR_002285 | https://fiji.sc/ |
| Software, algorithm | GraphPad Prism version 9.20 (283) for macOS | GraphPad Software, San Diego, CA | RRID:SCR_002798 | https://www.graphpad.com/ |
| Software, algorithm | Adobe Illustrator CC | Adobe Systems Inc | RRID:SCR_010279 | |

**Appendix 1—table 1.** Replicated, anonymized brood size assay (Dowen Lab).

| Strain | Tolkin (v1) | Tolkin (revised) | Li (submitted) | Dowen (anon.) | Li vs. Dowen |
|---|---|---|---|---|---|
| NK2571 (*qy78; cpIs122*) | 155±24.4 (n=19) | 212.8±27.5 (n=23) | 220±41 (n=15) | 233±32 (n=11) | t=0.899, p=0.378, n.s. |
| DG5020 (*lim-7p::ced-1::GFP; naIs37*) | 235.8±43.2 (n=56) | 237.5±46.5 (n=24) | 202±29 (n=12) | 213±52 (n=12) | t=0.643, p=0.527, n.s. |

