## [Editor Report]

This work extends the previous findings by the authors that suggested the lack of 'bare region' in the *C. elegans* gonad, which was previously postulated to exist between germ cells that are encapsulated by the distal tip cell and those that are encapsulated by sheath cells. The authors addressed the concerns posed by Tolkin et al. that proposed that the bare region does exist. However, discrepancies remain between the current manuscript and the manuscript by Tolkin et al., which should be resolved in the field in the future. Overall, the work presented here is important and of broad interest as it concerns the regulation of the stem cell niche and how cells that are destined to differentiate exit the niche and proceed to differentiation by interacting with the stromal cells.

---

## [Decision Letter]

**Decision letter after peer review:**

Thank you for submitting your article "The *C. elegans* gonadal sheath Sh1 cells extend asymmetrically over a differentiating germ cell population in the proliferative zone" for consideration by *eLife*. Your article has been reviewed by 3 peer reviewers, including X as the Reviewing Editor and Reviewer #1, and the evaluation has been overseen by Anna Akhmanova as the Senior Editor. The following individuals involved in review of your submission have agreed to reveal their identity: Judith Yanowitz (Reviewer #2); Ekaterina Voronina (Reviewer #3).

The reviewers agreed that the manuscript represents an important advance that is based on the previous *eLife* paper (Gordon et al.). However, there are important issues brought up by reviewers that need to be addressed as you can see below.

*Reviewer #1:*

Overall, it seems that this manuscript presents a compelling case that their earlier conclusion was correct. However, the data as well as explanation are sometimes not detailed enough, making it difficult to assess the rigor of the work. As detailed below, I'd like to see a bit more data that affirm their conclusion.

– line 62-69 is very difficult to understand. The emb-30 mutant and glp-1 mutant are not explained sufficiently (what they are, what phenotypes to expect in more details are needed), making it hard to understand the flow of the logic. The results seem to be consistent with the notion that Sh1 influences the differentiation of germ cells, once they exit the stem cell niche created by the DTC. But I think at least more thorough explanation is necessary, and perhaps additional more convincing data are needed. In Figure 1, 'germ cell phenotype' is only assessed by DAPI staining (the other markers only telling the location of DTC and Sh1) and it cannot be concluded for sure whether their assessment is correct.

– Fig4E (legend says it's G, please correct) - what it tries to show is unclear to non-expert. Please add control and elaborate exactly what is a defect.

– Figure 5: this is a critical figure to prove that Tolkin et al. might have used a combination of transgenic alleles that together causes germ cell defects. However, the 'phenotypes' is only assessed by DAPI in Figure 5, and it is unclear whether their observations (defects, or lack thereof) are comparable to what Tolkin et al. described in their manuscript.

*Reviewer #2:*

The Li et al. paper is essentially a response to recently submitted manuscript, Tolkin et al. 2021 and a follow-up on the senior author's prior work characterizing the relationship between Sh1, the DTC, and germ cell proliferation. Overall the manuscript provides evidence that the ced-1::GFP marker in Sh1 appears to be deleterious and therefore argues for caution when using extrachromosomal arrays and when relying heavily on single reporters to assess cell morphology. However, there is little new in the data, but rather it stands as a commentary to Tolkin et al.

One of the critical issues here as that the Tolkin paper suggests that the tagged inx-8 allele should not be used as a Sh1 marker since it is dominant negative. This paper suggests that there are defects associated with lim-7::CED-1::GFP which preclude this as a marker. While substantive data supports the latter (albeit with fairly low N numbers), the authors do not address whether there might be problems with the inx-8 mKate strain. They do, however, argue about the validity of their observations by a detailed analysis of imaging data that accounts for weak fluorescence and by use of multiple difference promoter GFP transgenes.

One of the major areas of conflict between this paper and Tolkin et al. is whether the inx-8 tagged strains has brood size defects: this paper and prior work say no; the Tolkin paper says yes. Are there differences in the growth media or temperatures used in the different labs? This should be resolved between the labs.

There is some concern that a subset of the markers are extracellular vs intracellular and the extracellular markers could be cleaved – an issue that is dismissed.

For all figures, the transgenes/tagged proteins used and specific strains must be listed in the figure and legend.

The statement that "A recent study (Tolkin et al., 2021) uses this transgene in all of the backgrounds analyzed (sometimes detecting the CED-1::GFP by anti-GFP antibody staining, which appears to amplify the variability of the marker), so we find this problematic reagent to undermine that study's conclusions." Is untrue. The Tolkin paper also used Plim-7::GFP without CED-1. Please modify this statement and clarify how you think that marker would be incorrectly analyzed.

It is very difficult to interpret the images in Figure 1 and 2: in some cases the DTC has long projection, in others it does not; in some images the germ lines are fat (Figure 1A,B) or bent and deformed (Figure 2A,C). IN figure 2F, it is unclear why this is DIC +GFP since the full GFP signal may not be apparent with the DIC and why are there lots of eggs around. Overall an improvement in the imaging would better help the reader understand the data and compare across figures/genotypes.

The description of the emb-30 and glp-1 alleles is confusing. It is not clear why these alleles would be a test of their hypothesis. Also of concern is that the shift of these alleles is done is L4/adulthood after the Sh1 cells is differentiated? Are they arguing that the Sh1 does not change in morphology in response to the shift in the germ cell state?

In Figure 2A-C, an asterick should be used to mark the DTC.

In text it would be helpful to better describe what Figure 1 is showing. What do you mean by "pink arrow marking gc transition".

The actual distance here is irrelevant if the germ lines are different sizes. A better measure is the number of nuclear widths.

In the methods, it is stated that, 16{degree sign}C experiments are cultured longer (for emb-30), but what is the evidence to support the choice of a 3 hour difference? Why is a similar criterion not use for the glp-1(ts)?

DAPI staining in the methods: "DAPI staining was done according to standard protocols". Needs to be referenced. I am shocked at the concentration of DAPI. My understanding is that concentrations of 1/50,000 dilution of 10ug/ml stock are used. Was this really not in water and not buffer?

While the desire to be concise is appreciate, in the methods or an accompanying table, the full description of the strains must be provided.

There needs to be clarification as to the source of the strains. (perhaps in a strain table).

What is meant by "DG5020 and DG5131 were shipped overnight on 9/23". Shipped from where?

In table 1, subscript "b" says: "N2 numbers come from multiple trials, not all of which counted negligible numbers of dead embryos, including the trial in which ina-1(qy23) and cam-1(cp243) were counted". Does this mean that in a subset of the trials, there was embryonic lethality in the N2 strain? How is the accounted for? Why is it not shown?

Also related table 1, it is stated: "our endogenously tagged inx-8(qy78) and inx-9(qy79) strains had dramatically different degrees of embryonic lethality, with qy79 producing over 150 unhatched eggs per worm." When I look at the table, the brood size for inx-9 is ~226 with 41% embryonic lethality. This would lead to on average ~85 unhatched eggs/worm not 150. Where do these numbers come from and which is correct?

Given that there are differences between the labs in the behaviors of different strains, it is critical that Kacy et al. perform the control brood sizes on the inx-8 and inx-9 described in lines 231 -236 since a critical aspect of their interpretation hinges on this.

Lines 239-240, please provide references for other brood sizes conducted in the lab.

What do you mean in Figure 4 by aberrant engulfment? It seems it is just pointing to foci at the cell membranes that could be junctions between the cells.

Lines 114-121 should refer to an image figure (3C,D)

*Reviewer #3:*

In this manuscript, Li et al. describe their further investigation of somatic Sh1 cells association with underlying germ cells. This work provides further support for the model where the distal border of Sh2 cell is forming an interface with DTC and Sh1 overlays the mitotic germ cells poised for differentiation. These findings confirm and extend the previous report, but do not test the functional significance of Sh1/germ cell association. Additionally, through the analysis of coexpressing CED-1::GFP and mKate::INX-8, this work revealed that two Sh1 cells may have drastically different levels of CED-1::GFP expression, while maintaining relatively similar mKate::INX-8 levels. The distinct regulation of transgene expression and asymmetric architecture of Sh1 cells around the germ cells have not been appreciated before, but the functional significance of these findings is unclear. The authors argue that overexpression of CED-1::GFP is detrimental as it is associated with gonad morphology defects, and further propose that high levels of CED-1::GFP expression in distal Sh1 are not well-tolerated.

Overall, this manuscript presents findings that confirm the previously published model; however, several discrepancies between this manuscript and Tolkin et al. preprint are concerning:

1) bcIs39 CED-1::GFP transgene was reported to perfectly overlap with mKate::INX-8 in >85% of cases by Tolkin et al., but only in 60% of cases in the current work.

2) The reported brood size and embryonic lethality for several identical or equivalent strains are vastly different. Specifically, NK2571 inx-8(qy78) brood size is 220 vs 155 with embryonic lethality of 8% vs 58% and DG505 embryonic lethality is 20% vs 1%. The causes for these differences need to be identified (culture conditions? temperature? formulation of culture plates?); otherwise these problems will persist/emerge for any other research group using these markers leading to issues with result reproducibility. In contrast to the assertion in Line 243, Tolkin et al. have assessed qy78 allele for brood size and embryonic lethality by itself after an outcross or in the NK2571 strain containing cpIs122 transgene, not in combination with bcIs39. Therefore, one cannot argue that brood size and embryonic lethality defects originated from the genetic interaction of qy78 and bcIs39.

Additionally, a detailed analysis of CED-1::GFP marker of Sh1 in Figure 4 revealed that ~50% of morphologically-normal gonads display an interface between DTC and Sh1, while remaining morphologically-normal gonads show a gap between these cells. This provides an opportunity to test the assertion of the model put forth by Gordon et al. 2020 and challenged by Tolkin et al. preprint – that the distal boundary of Sh1 cells impacts germ cell switch from proliferation to differentiation. According to Gordon et al., 2020 model, the proximal displacement of Sh1 in 50% of gonads expressing CED-1::GFP is expected to shift the position of meiotic entry away from the distal end resulting in a larger distal mitotic region in these germlines. By contrast, data in Figure 5 shows a shorter mitotic region in both strains expressing CED-1::GFP, consistent with Tolkin et al's conclusion that Sh1 position does not affect meiotic entry. Therefore, it appears that while the normal position of Sh1 distal boundary is closer to DTC than previously appreciated, its displacement is unlikely to affect the underlying germ cell population.

Other suggested revisions:

1. Line 180: the conclusion that bcIs39 "sensitizes worms for gonad morphology defects" is unwarranted as disruption of DTC migration appears similar in both described genetic backgrounds. It appears that bcIs39 directly disrupts DTC migration.

2. Line 196: remove "of".

3. Line 209: using CRISPR/Cas9 *to* introduce… (add "to").

4. Figure 1 and legend: include the allele designations of edited inx-8 and inx-9 for consistency with other figures.

5. Figure 1D: The position of Sh1 distal boundary in the right column (restrictive temperature) is hard to judge; the dashed line indicates a distal projection in the middle that is not apparent by diffuse GFP signal.

6. line 546, 557, 564 and Figure 1E: I don't think "gc transition" is an accepted term in the field. Perhaps replace with "mitotic cell population boundary"?

7. line 571: the legend indicates strain ID only for DG5020; is this necessary? If so, all strain IDs need to be included.

8. line 599: change panel to (D).

9. line 609: change panel to (E).

---

## [Author Response]

Reviewer #1:Overall, it seems that this manuscript presents a compelling case that their earlier conclusion was correct. However, the data as well as explanation are sometimes not detailed enough, making it difficult to assess the rigor of the work. As detailed below, I'd like to see a bit more data that affirm their conclusion.-line 62-69 is very difficult to understand. The emb-30 mutant and glp-1 mutant are not explained sufficiently (what they are, what phenotypes to expect in more details are needed), making it hard to understand the flow of the logic. The results seem to be consistent with the notion that Sh1 influences the differentiation of germ cells, once they exit the stem cell niche created by the DTC. But I think at least more thorough explanation is necessary, and perhaps additional more convincing data are needed. In Figure 1, 'germ cell phenotype' is only assessed by DAPI staining (the other markers only telling the location of DTC and Sh1) and it cannot be concluded for sure whether their assessment is correct.

Thank you very much for pointing out these experiments are not described sufficiently well. We have expanded the description of what these temperature sensitive alleles have been used for in the past, how our results agree with these prior findings, and furthermore how our result reveal the position of the sheath relative to the stem-like population of germ cells. Pages 1-3 have a rewritten section describing these experiments.

This assay is not demonstrating that Sh1 influences the germ cells. Instead, the original experiments helped to identify a feature of the germline—its division into a distal, stem like population and a more proximal, mitotically proliferating population that is fated to differentiate though has not yet entered the meiotic cell cycle. We find that the distal position of Sh1 falls at the same proximodistal position as the boundary between stem and non-stem germ cells, and it is the cells beneath Sh1 that differentiate first when progress through the cell cycle is halted or the germ cell receptor of the stemness cue is deactivated. This evidence independently supports the conclusion we drew from the experiments reported in Gordon et al., 2020, Figure 5.

While we are eager to share the results of an experiment that builds of off foundational work in the field to confirm our 2020 findings, we did these experiments independently of and prior to our engagement with the Tolkin et al. group, and they are not meant to address that study directly. If our paper ends up published as a direct response to Tolkin et al., we are open to removing Figure 1 to keep the study focused. We appreciate your guidance on this subject.

– Fig4E (legend says it's G, please correct) - what it tries to show is unclear to non-expert. Please add control and elaborate exactly what is a defect.

Thank you for pointing this out; we agree that this figure needs more explanation (and a correct letter in the legend!). The lim-7p::CED-1::GFP transgene is typically used to visualize the engulfment of apoptotic germ cells near the bend of the gonad; this region of the germline undergoes physiological apoptosis, in which dying germ cells give cytoplasm to their sisters to inflate oocytes. The CED-1::GFP signal on the sheath membrane forms concentrated bubbles around germ cells being engulfed. We observe such “bubbles” of GFP in some samples of DG5131 (2/18 in the relevant dataset) that are located abnormally distal in the gonad (2 projections of the same gonad from Figure 4E). There are so many abnormal Sh1 membrane protrusions that we agree with the reviewer that the full projection is difficult to understand. This isn’t a phenotype we want to characterize, but it provides further evidence that this strain has abnormal gonad biology.

We have updated the projection shown in in Figure 4E and labeled the engulfed germ cells.

– Figure 5: this is a critical figure to prove that Tolkin et al. might have used a combination of transgenic alleles that together causes germ cell defects. However, the 'phenotypes' is only assessed by DAPI in Figure 5, and it is unclear whether their observations (defects, or lack thereof) are comparable to what Tolkin et al. described in their manuscript.

Thank you for the helpful suggestion to compare our results directly with the measurements reported by Tolkin et al. The most relevant figure from Tolkin et al. to the results shown in our Figure 5 is the experiment in Tolkin Figure 4, in which fixed, dissected gonads of various genotypes were antibody stained for the CYE-1(+) progenitor pool and anti-GFP against the lim-7p::ced-1::GFP transgenic protein that they use to label the sheath. The CYE-1(+) technique (from Mohammad et al., 2018) reveals a proximal boundary of CYE-1(+) cells that falls a bit distal to the crescent shaped nuclei of the “Transition Zone”. Therefore, while we don’t expect the PZ measurements in each dataset to be an exact match (ours should be longer), we do expect the magnitude and direction of difference between strains to be concordant*.*

We can use Mohammed et al. (2018) Figure 2 for the wild-type position of a CYE-1(+) domain. The Tolkin analysis lacks a wild-type control and does not analyze the *qy78* allele on its own without *bcIs39(lim-7p::ced-1::gfp)* in the background (“NA” in Author response table 1). Also, without the raw data, I must infer the mean measures from the display items in Mohammed Figure 2, Tolkin Figure 4, and its supplement. With those caveats, this is my best effort to compare between the experiments:

**Author response table 1. sa2table1:** 

Strain, Genotype	Tolkin Figure 4 label	Tolkin CYE-1(+) domain	Mohammed Figure 2 CYE-1(+) domain	Mohammed Figure 2 DAPI, est. from 1 sample	Li Figure 5 label	Li DAPI mean measurement of PZ
N2 wild-type	NA	NA	~95 um~21 gcd	~113 um~28 gcd	Wild type N2	109 +/- 13 um27 +/- 3 gcd
NK2571 (*qy78;cpIs122)*	NA	NA	NA	NA	*inx-8(qy78);cpIs122*	106 +/- 12 um26 +/- 2 gcd
DG5020 (*bcIs39; naIs37)*	“markers only” (4A)“Wild-type” (4C)	~70 um~17 gcd	NA	NA	DG5020(*bcIs39, naIs37*)	91 +/- 19 um23 +/- 3 gcd
DG5131 *(qy78;bcIs39; naIs37)*	inx-8(qy78)	~60 um~15 gcd	NA	NA	DG5131(*qy78;bcIs39; naIs37*)	79 +/- 13 um20 +/- 2 gcd

First, as expected, when both measurements have been made for the same strain, the CYE(+) domain is ~5-6 germ cell diameters or ~20 microns shorter than the DAPI-stained Proliferative Zone. This agrees with the observations of Mohammed et al. (2018). Next, wild-type N2 have the longest proliferative zone, with DG5020 having a somewhat shorter proliferative zone, and DG5131 having a profoundly shorter proliferative zone. In our analysis, we find that NK2571(*qy78;cpIs122*) has a proliferative zone that is statistically indistinguishable from wild type (Tolkin did not analyze either fully wild-type or NK2571 in this experiment). Importantly, the results shown in Tolkin Figure 4 and its supplement show that DG5020 has an abnormal distal gonad compared to previously published wild-type measurements, which is consistent with our findings.

We have now updated our Results section in Line 281-323 to clarify this comparison. We also have increased the N for all strains analyzed (see Figure 5 legend).

To address the issue that we report only lengths of the proliferative zone using DAPI staining, in this revision, we add two additional measurements: incidence of germ cell division and total gonad length (Figure 5B and 5E). Overall, these tell a consistent story: in the NK2571 strain for which we observe a far distal Sh1 cell boundary, the germline is statistically indistinguishable from wild type*.* On the other hand, the “markers only” DG5020 strain preferred by Tolkin et al. is slightly diminished in all measures. Finally and crucially, the combination of alleles in the DG5131 strain that Tolkin uses to assess the function of the *qy78* allele in nearly all experiments is dramatically impaired both for the size of the proliferative zone, the number of actively dividing germ cells observed, and the length of the gonad. We believe this is strong evidence to refute the idea that *qy78* is a “poison” allele on its own; instead, there is a synthetic interaction with the markers used by Tolkin et al.

Reviewer #2:The Li et al. paper is essentially a response to recently submitted manuscript, Tolkin et al. 2021 and a follow-up on the senior author's prior work characterizing the relationship between Sh1, the DTC, and germ cell proliferation. Overall the manuscript provides evidence that the ced-1::GFP marker in Sh1 appears to be deleterious and therefore argues for caution when using extrachromosomal arrays and when relying heavily on single reporters to assess cell morphology. However, there is little new in the data, but rather it stands as a commentary to Tolkin et al.One of the critical issues here as that the Tolkin paper suggests that the tagged inx-8 allele should not be used as a Sh1 marker since it is dominant negative. This paper suggests that there are defects associated with lim-7::CED-1::GFP which preclude this as a marker. While substantive data supports the latter (albeit with fairly low N numbers), the authors do not address whether there might be problems with the inx-8 mKate strain.

Thank you for this suggestion to improve our sample sizes and to draw attention to the discussion of the shortcomings of our own fluorescently marked strains.

We have made our discussion of these weaknesses explicit (lines 418-421):

“In the end, no transgenic or genome-edited strain is truly wild type, and it should be our expectation that such strains might be somewhat sensitized as a result. Indeed, the synthetic interaction we document between *lim-7p::ced-1::gfp* and *inx-8(qy78)* suggests that the *qy78* is sensitized for gonad defects caused by other genetic elements affecting the gonadal sheath.”

We report and discuss the reduced brood sizes of *qy78* and other fluorescently marked strains in Table 1 and discussion thereof.

We have also increased our sample sizes (in all figure legends).

They do, however, argue about the validity of their observations by a detailed analysis of imaging data that accounts for weak fluorescence and by use of multiple difference promoter GFP transgenes.

We appreciate that the reviewer found the congruence between different fluorescence patterns to be good evidence for the existence of a distal Sh1 boundary. To clarify, these are not promoter::GFP transgenes in Figure 2A-D, these are CRISPR/Cas9-mediated fusions between endogenous protein coding genes and fluorescent protein coding genes. In addition to the *inx-8* and *inx-9* tags, we also observe distal Sh1 expression in CRISPR-tagged *ina-1* and *cam-1* strains. These latter two genes are not innexin genes (one encodes an integrin subunit, the other a Wnt receptor complex member). These proteins localize to the cell membrane.

The fluorescence in Figure 2A-D is dimmer and more punctate because these are tagged worm proteins expressed by their two genomic copies from endogenous promoters, localizing to the cell membranes in particular ways. This makes them less variable and less prone to silencing than overexpressed transgenes. However (related to the prior point about *qy78*), these tags also may potentially affect the function of the tagged proteins themselves. We consider this risk, and ways to address it, in the Discussion section (lines 381-416). We do not see a way in which tagging integrin or a Wnt receptor could affect innexin hemichannel function in the same way, however, so the distal Sh1 boundary is not only seen in strains with abnormal innexin proteins, as Tolkin et al. propose.

One of the major areas of conflict between this paper and Tolkin et al. is whether the inx-8 tagged strains has brood size defects: this paper and prior work say no; the Tolkin paper says yes. Are there differences in the growth media or temperatures used in the different labs? This should be resolved between the labs.

We agree that this crucial question must be resolved, and it was our priority in revising the manuscript (see first section of this letter). The Tolkin paper has now dropped the brood size defect and embryonic lethality claims.

There is some concern that a subset of the markers are extracellular vs intracellular and the extracellular markers could be cleaved – an issue that is dismissed.

Based on 3D models of another innexin (Oshima et al., 2016), we suspect our N-terminal tags on mKate::INX-8 and GFP::INX-9 are extracellular. We do not address (or explicitly dismiss) potential N-terminal cleavage in this manuscript, but we also don’t suspect it for several reasons. First, the mKate::INX-8 fluorescence is punctate, suggesting that anchoring of some kind rather than free diffusion within the gonad governs its localization. Second, where mKate::INX-8 is coexpressed with a DTC or sheath membrane marker (lag-2p::mNG::PH or lim-7p::CED-1::GFP), we see mKate colocalization at the cell membrane, not diffusion away from the GFP signal. Finally, we see considerable localization of the mKate::INX-8 signal within oocytes and embryos (Gordon et al., 2020, Figure S1), which Starich et al. 2014 report in their antibody staining experiments (that paper explains that the INX-8 protein is internalized from the sheath-germ cell interface, not produced by the embryos or germ cells). While this question of protein processing is interesting, we worry it would be a distraction to include this discussion in the manuscript. Our extracellular, membrane-tethered fluorescent protein tags might circumvent a problem with visualizing the Sh1 cell, which is that its thin, flat shape leaves relatively little room for cytoplasmic fluorescent proteins to accumulate to visible levels, hence the “honeycomb” appearance of cytoplasmic GFP in that cell.

In the revised version of the manuscript, Tolkin et al. add a CRISPR tagged FASN-1::GFP as well, a GFP fusion to an endogenous locus encoding fatty acid synthase (presumably with cytoplasmic localization). They have also added an extrachromosomal array transgene expressing a fusion of ACY-4::GFP, which is predicted to be localized to the membrane, and which has the most “Class 2” gonads with a DTC-Sh1 interface of any of their markers (Figure 1—Figure supplement 1). The same caveats about variability of extrachromosomal arrays apply to this strain; absence of expression should not be construed as absence of a cell.

For all figures, the transgenes/tagged proteins used and specific strains must be listed in the figure and legend.

Thank you for this suggestion. We hope to keep the nomenclature transparent and accessible, but agree with the reviewer that precision is very important and now include full strain information in all figure legends.

The statement that "A recent study (Tolkin et al., 2021) uses this transgene in all of the backgrounds analyzed (sometimes detecting the CED-1::GFP by anti-GFP antibody staining, which appears to amplify the variability of the marker), so we find this problematic reagent to undermine that study's conclusions." Is untrue. The Tolkin paper also used Plim-7::GFP without CED-1. Please modify this statement and clarify how you think that marker would be incorrectly analyzed.

Thank you for pointing out the need to correct “all” to “most” (line 396). Tolkin et al. do indeed use *lim-7* promoter-driven cytoplasmic GFP strains in Figure 1—Figure Supplement 1 (where the *lim-7p::GFP* is coexpressed with a DTC marker in otherwise wild-type and *inx-14(ag17*) backgrounds) and Figure 2—Figure Supplement 2 (where it is coexpressed with a DTC marker and *inx-8(qy78)*).

Our results for *lim-7::GFP* in Li Figure 2F do not agree with the results shown in Tolkin Figure 1—Figure Supplement 1; we find a mean distance of Sh1 from the distal end of 40.63 microns (N=20), Tolkin et al. show a mean of ~80 microns. Of note: this distal Sh1 position is dramatically different from what Tolkin et al. report for the *bcIs39[lim-7p::ced-1::gfp]* sheath marker strain (~50 microns, same figure). The internal inconsistency between these two markers is not discussed in that study.

We cannot comment further on this discrepancy except to say we measured sheath position in a different strain, strain DG1575 from the CGC (ordered 06-15-2020) which carries the *tnIs6* allele and was used to visualize the sheath in Hall et al., 1999. As for how it could be analyzed incorrectly, we suspect the same issue of variable/low levels of GFP expression from the *lim-7* promoter in the distal sheath could be at work in this case as well as for the *lim-7::ced-1::GFP* allele which we discuss at length.

We also analyzed this *tnIs6* allele in Gordon et al., 2020 (Figure 1—Figure Supplement 1C) in the presence of the *inx-8(qy78)* allele, and in that case find results that almost precisely agree with Tolkin’s conclusion in Figure 2—Figure Supplement 2. The two markers largely overlap (in our hands, 11/12 and 24/28 gonads in two separate replicates), and when they don’t, the mKate::INX-8 expression is farther distal than the GFP signal, due to apparent loss of GFP in the distal-most Sh1 cell.

It is very difficult to interpret the images in Figure 1 and 2: in some cases the DTC has long projection, in others it does not; in some images the germ lines are fat (Figure 1A,B) or bent and deformed (Figure 2A,C). IN figure 2F, it is unclear why this is DIC +GFP since the full GFP signal may not be apparent with the DIC and why are there lots of eggs around. Overall, an improvement in the imaging would better help the reader understand the data and compare across figures/genotypes.

Thank you for the suggestions to improve these figures. Asterisks have been added to the distal end of all gonads in every figure.

In Figure 1, we are not visualizing dedicated DTC markers. Fluorescence from the *cpIs122 lag-2p::mNG::PH* is quenched by methanol fixation for DAPI staining in Figure 1B (hence no DTC outline indicated), while the *inx-9(qy79)* allele has robust but punctate GFP::INX-9 localization on both Sh1 and DTC (outlined in white, Figure 1D). Additionally, we have now further highlighted that the experiments in Figure 1 depict older animals than the other figures, based on the timing used in the experiments we are replicating. The *emb-30* allele certainly causes gonads to look fat at the restrictive temperature; in this strain, germ cells stop moving proximally.

In Figure 2D and 2F, we feel it is important to show the DIC channel so the distal tip of the gonad is pictured. We have taken care not to wash out any GFP signal in the merged image. Eggs are sometimes visible, as the uterus lies immediately ventral to the distal tip of the gonad, and we are looking at reproductive adults in their first day of egg laying. Slight bends at the distal end of gonads are not deformities, but artifacts of coverslipping; the gonad tip is free in the body and when compressed, settles around gut or eggs nearby.

The description of the emb-30 and glp-1 alleles is confusing. It is not clear why these alleles would be a test of their hypothesis. Also of concern is that the shift of these alleles is done is L4/adulthood after the Sh1 cells is differentiated? Are they arguing that the Sh1 does not change in morphology in response to the shift in the germ cell state?

The reviewer is correct that our results show that the position of Sh1 does not change in these temperature sensitive mutants, and that its distal position does not differ from the position of the germ cell population that differentiates first after temperature shift. We have improved our explanation of these experiments thanks to this helpful feedback and the similar suggestions of Reviewer 1 (above). Specifically, we have explained that we are using the germ cell perturbations of the ts alleles to reveal distal (stem-like) and proximal (still mitotic, but more genetically differentiated) germ cell populations, in relation to the position of the Sh1 cells. In both experiments, we find that the pool of germ cells that differentiates first at the restrictive temperature is the pool that lies under the Sh1 cells. This confirms that these cells have left the niche.

In Figure 2A-C, an asterick should be used to mark the DTC.In text it would be helpful to better describe what Figure 1 is showing, What do you mean by "pink arrow marking gc transition".

The “transition zone” of the germline is the position where the nuclei take on the characteristic crescent shape of leptotene/zygotene of meiosis I. While some of the small, circular nuclei distal to this position are in S phase of meiosis I, the vast majority are mitotic germ cells, so the gonad distal to the transition zone is referred to as the proliferative zone. However, in the temperature-shifted experiments, it seemed wrong to call this the “proliferative zone” because the cells have ceased to proliferate. However, the difference in nuclear morphology can still be observed.

The actual distance here is irrelevant if the germ lines are different sizes. A better measure is the number of nuclear widths.

The results from the *emb-30* experiment are difficult to analyze in terms of germ cell diameters, because the germ cell nuclei are malformed due the failure of metaphase-anaphase transition. However, we added this measurement for one of the replicates (shown in Figure 1—Figure supplement 1E).

In the methods, it is stated that, 16{degree sign}C experiments are cultured longer (for emb-30), but what is the evidence to support the choice of a 3 hour difference? Why is a similar criterion not use for the glp-1(ts)?

Thank you for pointing out the need to better explain these experiments, which Reviewer 1 also noted (see above). These experiments were inspired by key prior work that provides evidence for a distal pool of stem-like germ cells; in this paper we repeat these experiments in the presence of our fluorescent DTC and sheath markers to observe the positions of the soma and stem-like germ cell pool. These genetic loss of function experiments provide corroborating evidence that the germ cells that lie under the Sh1 cells are further along the path to differentiation than the distalmost stem-like cells. Our 2020 study used fluorescent markers related to germ cell fate to first test this hypothesis.

The original *emb-30(ts)* experiment by Cinquin et al. (2010) imaged a starting point control but had no aged control that was kept at the permissive temperature while the experimental group was shifted to the restrictive temperature for 15 h. This seemed like a nonideal control to us since the gonad and germline change over time, so the initial 18 hour control was chosen for experimenter convenience; the experimental group was fixed, stained, and imaged at 15 hours with the control continuing to run in the meantime. We have since repeated the experiment with a starting point control only (as in the original study), and we see the same result that we obtained with the 18 hour control, as well as a 21 hour control we recently performed (Figure 1—Figure supplement 1). This suggests that by 36 hours post mid-L4, the relationship between the stem cell zone and Sh1 cell are fixed.

On the other hand, the original *glp-1(bn18)* experiments of Fox and Schedl (2015) included controls that were exactly matched for time, so our use of this allele similarly uses a control kept at the permissive temperature for the same amount of time that the experimental group is observed.

DAPI staining in the methods: "DAPI staining was done according to standard protocols". Needs to be referenced. I am shocked at the concentration of DAPI. My understanding is that concentrations of 1/50,000 dilution of 10ug/ml stock are used. Was this really not in water and not buffer?

Thank you for pointing this out! DAPI was reconstituted in distilled water per the MSDS, but you are correct that the working solution was made in 0.01% Tween in PBS. The lower end of suggested working concentrations given on the MSDS is 1 ug/ml. We noted that this is higher than the Nayack Lab protocol suggests, but we get clean staining, even if we waste some dye. Since we cut our methanol fixation short to preserve mKate::INX-8 fluorescence, we probably have lower tissue penetration to compensate for.

While the desire to be concise is appreciate, in the methods or an accompanying table, the full description of the strains must be provided.

Thank you, a Key Resources Table has now been composed. We have also improved Table 1 as suggested (below).

There needs to be clarification as to the source of the strains. (perhaps in a strain table).What is meant by "DG5020 and DG5131 were shipped overnight on 9/23". Shipped from where?

The full details for each strain now appear in the Key Resources Table.

Perhaps that level of detail on strain history is unnecessary. We wished to express that the strains we imaged for this manuscript were analyzed directly after receipt from the Tolkin et al. collaborators, not after long-term culture (during which strains can theoretically diverge). We understand that this is not a typical part of a strain description, but when we are looking at nominally the same strains and reporting different phenotypes, it seems important. We did however delete references to details of strain maintenance in the Results section.

In table 1, subscript "b" says: "N2 numbers come from multiple trials, not all of which counted negligible numbers of dead embryos, including the trial in which ina-1(qy23) and cam-1(cp243) were counted". Does this mean that in a subset of the trials, there was embryonic lethality in the N2 strain? How is the accounted for? Why is it not shown?

We have not typically quantified embryonic lethality when performing brood size assays, and we always include an N2 wild-type control population. In this case, we ran one brood size assay for strains LP520 and NK2324 with a wild-type N2 control and did not score embryonic lethality. Later, when we received the Tolkin strains, we ran a brood size assay for DG5020 and DG5131 along with our strains NK2571 and KLG019 and an N2 control, and we did score embryonic lethality because we saw that the other group had done so. So while we can pool the N2 brood size, we can’t report a total embryonic lethality for the same dataset.

Also related table 1, it is stated: "our endogenously tagged inx-8(qy78) and inx-9(qy79) strains had dramatically different degrees of embryonic lethality, with qy79 producing over 150 unhatched eggs per worm." When I look at the table, the brood size for inx-9 is ~226 with 41% embryonic lethality. This would lead to on average ~85 unhatched eggs/worm not 150. Where do these numbers come from and which is correct?

We constructed our table in as similar a fashion to Tolkin et al., which reports *live* brood only in the brood size column as a mean and standard deviation of total live offspring, and report embryonic lethality as a percentage of all offspring (live worms+unhatched eggs), though the total numbers of unhatched eggs is never presented. We have now presented our findings differently in the revised version of the manuscript, with total number of unhatched eggs (as well as percent embryonic lethality) listed alongside live brood.

Given that there are differences between the labs in the behaviors of different strains, it is critical that Kacy et al. perform the control brood sizes on the inx-8 and inx-9 described in lines 231 -236 since a critical aspect of their interpretation hinges on this.

Thank you for pointing out the need to clarify this text. The text in line 231-236 cites prior work on a different set of innexin mutants (Starich and Greenstein, 2020 https://doi.org/10.3390/biom10121655), which refers to broods in the mid-200s as “nearly wild-type”. We referred to this qualitative assessment to contextualize the brood sizes of the transgenic strains in our analysis. We agree with the reviewer that introducing discussion of these other *inx-8* mutants is distracting. The text now reads in lines 275-277:

“All of the fluorescently marked strains have mildly to moderately reduced brood sizes. On the basis of brood size alone, there is not a strong reason to prefer one of these markers over another.”

Lines 239-240, please provide references for other brood sizes conducted in the lab.

This line referenced unpublished observations in our lab and previously in the Sherwood lab and has been removed. The numbers Tolkin et al. report—a brood size of 108 with Day 1 embryonic lethality of 87.7%--would be impossible to miss during routine culture and would make performing crosses with this allele difficult. We have seen no such brood size defect in our years working with the strain.Tolkin et al. now drop this claim.

What do you mean in Figure 4 by aberrant engulfment? It seems it is just pointing to foci at the cell membranes that could be junctions between the cells.

Thank you for pointing out the need to better describe this phenotype; Reviewer 1 had the same question, which we address above in the figure included in this letter.

Lines 114-121 should refer to an image figure (3C,D).

Call out to figure 3A has been added, thank you!

Reviewer #3:In this manuscript, Li et al. describe their further investigation of somatic Sh1 cells association with underlying germ cells. This work provides further support for the model where the distal border of Sh2 cell is forming an interface with DTC and Sh1 overlays the mitotic germ cells poised for differentiation. These findings confirm and extend the previous report, but do not test the functional significance of Sh1/germ cell association. Additionally, through the analysis of coexpressing CED-1::GFP and mKate::INX-8, this work revealed that two Sh1 cells may have drastically different levels of CED-1::GFP expression, while maintaining relatively similar mKate::INX-8 levels. The distinct regulation of transgene expression and asymmetric architecture of Sh1 cells around the germ cells have not been appreciated before, but the functional significance of these findings is unclear. The authors argue that overexpression of CED-1::GFP is detrimental as it is associated with gonad morphology defects, and further propose that high levels of CED-1::GFP expression in distal Sh1 are not well-tolerated.Overall, this manuscript presents findings that confirm the previously published model; however, several discrepancies between this manuscript and Tolkin et al. preprint are concerning:1) bcIs39 CED-1::GFP transgene was reported to perfectly overlap with mKate::INX-8 in >85% of cases by Tolkin et al., but only in 60% of cases in the current work.

The “>85%” figure from Tolkin et al., Figure 2 Supplement 2B is derived from 11/13 gonads where the two markers overlap. The image they select to illustrate an example of nonoverlapping expression indeed does show overlapping expression, however, they have just underexposed or underscaled the GFP signal in the distal portion of the sheath (see visible Sh1 features). This observation informed our approach, and instead of looking for cases of “non overlapping expression” we counted cases where two vastly different GFP expression intensities in the two Sh1 cells could lead to a similar inadvertent mis-scaling. We report discrepancies in the coexpressed markers in 6/15 cases in a dataset that was collected for this particular experiment (imaging through the full thickness of the gonad with long exposure times to see the entirety of both Sh1 cells). In other DG5131 datasets, we see discrepancies in expression levels between cells in 5/10 gonads (imaged by Singh on 10/8) and 7/16 gonads (imaged by Gordon, 10/5).

2) The reported brood size and embryonic lethality for several identical or equivalent strains are vastly different. Specifically, NK2571 inx-8(qy78) brood size is 220 vs 155 with embryonic lethality of 8% vs 58% and DG505 embryonic lethality is 20% vs 1%. The causes for these differences need to be identified (culture conditions? temperature? formulation of culture plates?); otherwise these problems will persist/emerge for any other research group using these markers leading to issues with result reproducibility. In contrast to the assertion in Line 243, Tolkin et al. have assessed qy78 allele for brood size and embryonic lethality by itself after an outcross or in the NK2571 strain containing cpIs122 transgene, not in combination with bcIs39. Therefore, one cannot argue that brood size and embryonic lethality defects originated from the genetic interaction of qy78 and bcIs39.

Tolkin et al. have now dropped those criticisms. However, because the Tolkin paper used “wild type” to refer to their strain carrying the bcIs39 marker and DTC marker, we were unsure about the precise genotypes they measured brood size for. Unfortunately, we did not have access to the strains that Tolkin et al. refer to as “qy78” in their table, which are described as a 1x and 2x outcross of our NK2571 strain to “wild-type males”. We recapitulated this experiment by outcrossing to wild-type N2 males, and again we see no dramatic embryonic lethality phenotype caused by the *qy78* allele (see first section of letter).

Additionally, a detailed analysis of CED-1::GFP marker of Sh1 in Figure 4 revealed that ~50% of morphologically-normal gonads display an interface between DTC and Sh1, while remaining morphologically-normal gonads show a gap between these cells. This provides an opportunity to test the assertion of the model put forth by Gordon et al. 2020 and challenged by Tolkin et al. preprint – that the distal boundary of Sh1 cells impacts germ cell switch from proliferation to differentiation. According to Gordon et al., 2020 model, the proximal displacement of Sh1 in 50% of gonads expressing CED-1::GFP is expected to shift the position of meiotic entry away from the distal end resulting in a larger distal mitotic region in these germlines. By contrast, data in Figure 5 shows a shorter mitotic region in both strains expressing CED-1::GFP, consistent with Tolkin et al's conclusion that Sh1 position does not affect meiotic entry. Therefore, it appears that while the normal position of Sh1 distal boundary is closer to DTC than previously appreciated, its displacement is unlikely to affect the underlying germ cell population.

In the past we had hoped we could use some of these transgene overexpression strains (specifically our *lim-7p::gfp::caax* transgenic) to perturb the sheath position, however we strongly suspect that these strains have a failure to label the distal Sh1 cell rather than a displacement of the distal Sh1 cell away from the distal end. We have no way to know for sure given these tools. After observing so many transgenic animals with the GFP patterns observed in Figure 2H and Figure 3B’—and especially 3D—I don’t feel confident concluding that the Sh1 cell is *not* there, based on absence of transgene fluorescence alone. Absence of evidence is not evidence of absence, as they say. We discuss this in the results, lines 153-174.

However, we did displace the sheath from the distal end by applying RNAi against genes that encode key regulators of branched actin dynamics (*arx-2* and *unc-60*, which encode an Arp2/3 subunit and cofilin, respectively), we don’t shift the proliferative zone as recognized by nuclear morphology, but do shift the region in which GLD-1::GFP accumulates in germ cells (Gordon et al., 2020, Figure 7). We know that Sh1 is not necessary for differentiation (Killian and Hubbard, 2005), but that Sh1-ablated animals are temporally delayed for meiotic entry.

We suspect that the shorter mitotic regions in DG5131 CED-1::GFP;mKate::INX-8 (and to a lesser extent in DG5020 CED-1::GFP) animals reflect impaired germ cell proliferation due to Sh1 abnormality. Recent work that is not affiliated with our group (Gopal et al., 2021) finds that loss of syndecan in *sdn-1* mutant alleles and caused by *sdn-1* RNAi applied across the late larval-young adult transition shortens the mitotic region of the adult germline and impairs germ cell proliferation; the mutants are rescued by sheath specific expression of *sdn-1* (neither DTC nor germ cell expression rescues). Gopal et al. go on to demonstrate that the syndecan phenotype is a *glp-1*-mediated effect, implicating signaling from Sh1 on the Notch-dependent mitotic germline. This link is easily explained by the morphological overlap of Sh1 and the mitotic germline that Gordon et al. (2020) and Li et al. (under review) describe, and it is harder to explain if Sh1 does not contact the mitotic germline in adults, as Tolkin et al. claim.

Other suggested revisions:1. Line 180: the conclusion that bcIs39 "sensitizes worms for gonad morphology defects" is unwarranted as disruption of DTC migration appears similar in both described genetic backgrounds. It appears that bcIs39 directly disrupts DTC migration.

Thank you for this suggestion; the text (line 219-220) now reads: The *lim-7p::ced-1::GFP* transgene seems to cause incompletely penetrant gonad morphology defects.

2. Line 196: remove "of".3. Line 209: using CRISPR/Cas9 *to* introduce… (add "to").4. Figure 1 and legend: include the allele designations of edited inx-8 and inx-9 for consistency with other figures.

Thank you! Strains and alleles given.

5. Figure 1D: The position of Sh1 distal boundary in the right column (restrictive temperature) is hard to judge; the dashed line indicates a distal projection in the middle that is not apparent by diffuse GFP signal.

We agree that the merged image is a little difficult to read because of the strong germ cell nuclear expression, but the single channel image in the middle should make it clear that the Sh1 GFP expression forms a continuous sleeve over the distal germline. We mark all Sh1 distal edges with a dashed yellow line to help.

6. line 546, 557, 564 and Figure 1E: I don't think "gc transition" is an accepted term in the field. Perhaps replace with "mitotic cell population boundary"?

You’re right, we have updated to call it the “germ cell transition zone”. “Transition zone” is customary, and we clarify that this is a germ cell feature for those less familiar with the system.

7. line 571: the legend indicates strain ID only for DG5020; is this necessary? If so, all strain IDs need to be included.

Thank you, we agree that all strain ID should be given.

8. line 599: change panel to (D).9. line 609: change panel to (E).

Thank you very much for catching these!